# Multi-omics reveals the involvement of endophytes in the growth of Moso bamboo (*Phyllostachys edulis*) shoots

Aoshun Zhao [1,2], Manchang Huang[3], Yingjie Cheng[1], Qiaoling Li[1], Hanjiang Cai[1], Yufang Bi[1], Anke Wang[1], Xuhua Du [1] ✉ & Xingcui Ding [1,4] ✉

Moso bamboo (*Phyllostachys edulis*) exhibits extraordinary speed of growth. While its anatomical and hormonal features have been well studied, the contribution of microbial interactions to its rapid growth remains largely unknown. Here, we integrated 16S rRNA and ITS amplicon sequencing, phytohormone measurements, and root transcriptome analysis across four developmental stages and three plant compartments (shoot top, shoot bottom, and root). We show that microbial diversity and network complexity were strongly shaped by tissue type and developmental stage, with early-stage roots and shoot meristems exhibiting strong immune filtering and later stages showing a marked increase in diversity. Network analysis revealed highly complex microbial associations in nutrient-limited shoot tops during dormancy, suggesting influenced cooperation among endophytes. In roots, hormone levels were tightly correlated with microbial dynamics, and transcriptome analysis identified 153 hormone-related genes that are differentially expressed across developmental stages, including stage-specific activation of AUX/IAA and SAUR families. A plant-microbe-hormone interaction network highlighted associations between auxin-related genes and growth-promoting genera such as *Paenibacillus*. Together, these findings reveal that endophytes likely modulate hormone signaling to facilitate rapid shoot elongation, providing insights into the unique developmental program of Moso bamboo.

Plants play a crucial role in intricate and dynamically evolving ecosystems and influence their interactions with diverse microbial communities[1]. A rapid decline in microbial diversity from soil to roots and then to leaves and flowers, suggesting strong selective pressure along the soil-plant continuum[2]. To escape this selection pressure, microbial groups have evolved mechanisms to evade or overcome the host's immune system, including mimicking host signaling molecules, secreting immune-modulatory effectors, and forming protective biofilms within host tissues. These adaptations facilitate their successful colonization of plant compartments-such as roots, stems, or leaves-without inducing any pathogenic effects[3]. The assembly of endophytic microbial communities is influenced by the communication between microbes and plants[4]. The phytohormone network, composed of biosynthesis, transport, perception, and signaling pathways, plays a dominant role in coordinating plants and microbes interactions[5]. For example, plant-derived phytohormones can influence microbial metabolism and community assembly directly or indirectly[6]. However, microbes can also manipulate plant hormone networks by producing phytohormones such as indole-3-acetic acid, cytokinins and gibberellins, by secreting hormone mimics such as coronatine and phenazines that resemble plant hormones in structure, and by releasing proteinaceous effectors that alter host hormone biosynthesis or signaling[7]. Therefore, in plant-microbe interactions, the phytohormone network should be considered as an inter-kingdom signaling network integrating plant and microbial molecules and signals[8,9]. The specific mechanisms by which microbes participate in plant hormone signaling networks remain unclear.

Moso bamboo (*Phyllostachys edulis*) is recognized as the most ecologically and economically important bamboo species worldwide[10]. Moso bamboo enters a dormant phase during winter and breaks dormancy once favorable temperatures are reached, initiating rapid internode elongation

[1]China National Bamboo Research Center, Key Laboratory of State Forestry and Grassland Administration on Bamboo Forest Ecology and Resource Utilization, Hangzhou, Zhejiang, 310012, China. [2]Key Lab of Organic-Based Fertilizers of China and Jiangsu Provincial Key Lab for Solid Organic Waste Utilization, Nanjing Agricultural University, Nanjing, 210095, China. [3]College of Life Sciences, Xishuangbanna Vocational and Technical College, Jinghong, 666100 Yunnan, China. [4]Bamboo Industry Institute, Zhejiang A & F University, Hangzhou, Zhejiang, 311300, China. ✉e-mail: stary8@163.com; dxc01@hotmail.com

and accelerated shoot growth[11]. Moso shoots exhibit specialized structures adapted to their rapid growth, characterized by distinct differentiation of apical meristem and basal mature tissues[12]. Additionally, adventitious roots at the base of shoots display explosive growth[13,14]. Plant hormones play distinct roles in regulating the growth and development of different regions of Moso bamboo[13]. Gibberellin GA$_1$ accumulates at shoot tops, directly triggering its rapid growth, while abscisic acid (ABA) and salicylic acid (SA) predominantly accumulate at the bottom of shoots, inducing rapid stem thickening through the MYB83L pathway[14]. Thus, phytohormone signaling-associated genes play a vital role in the fast-growing shoots[11]. Previous studies have shown that endophytic microbes in Moso bamboo shoots significantly promote the growth of bamboo seedlings[15]. However, the mechanisms by which these microbes influence phytohormone networks to contribute to Moso bamboo development remain largely unexplored.

Although the anatomical and hormonal features of Moso bamboo shoots have been extensively studied, the contribution of endophytic communities to their exceptional growth rate remains unclear. Here, we profiled bacterial and fungal communities and quantified key phytohormones across three anatomical compartments (shoot top, shoot bottom, root) and four developmental stages, and conducted transcriptomic analysis of roots across the four developmental stages. We hypothesized that spatiotemporal shifts in endophytic community composition are coupled with hormone signalling changes in the root system, and that these microbe-hormone interactions play a regulatory role in the rapid shoot growth of Moso bamboo.

## Results

### Dynamics of microbial composition and structure of Moso shoots

To unveil the assembly patterns of root-shoot-associated microbiomes in Moso bamboo, we investigated microbiome dynamics across four developmental stages: dormant (S1), dormancy-breaking (S2), accelerated growth (S3), and rapid growth (S4). We studied three different compartments: the top of Moso bamboo shoot (shoot top, T), the bottom of Moso bamboo shoot (shoot bottom, B), the root of Moso bamboo (root, R) (Fig. 1A). Non-metric multidimensional scaling (NMDS) revealed that bacterial and fungal β-diversity were clearly separated among the different developmental stages and compartments of Moso shoots (Fig. 1B). Correspondingly, permutational multivariate analysis of variance (PERMA-NOVA) results suggested that plant stages were the main driver of bacterial β-diversity ($R^2 = 0.207$, $P < 0.001$). However, bacterial communities were clearly segregated in various plant compartments during S1(Fig. S2; $P < 0.001$). As Moso shoots were broken in dormancy (S2) and entered into the growth phase (S3, S4), the impact of different plant compartments on bacteria diversity progressively diminishes (Supplementary Fig. 1). Similarly, the impact of plant compartments on fungal community gradually declined (Supplementary Fig. 1).

The plant compartment, plant developmental stage exhibited significant effects on bacterial α-diversity (Chao1 index), with consistently higher α-diversity in the root than the shoots. As plants transitioned from S1 to S2, endophytic bacterial and fungal α-diversities significantly increased in roots (Fig. 1C). To validate the robustness of α-diversity patterns, additional indices (Shannon, ACE, Pielou) were also calculated, which showed similar trends (Supplementary Fig. 2).

The plant-associated bacteria mainly belonged to Gammaproteobacteria and Bacillales, with Bacillales predominant at the top of shoot and Gammaproteobacteria dominant at the bottom of shoots and roots. Intriguingly, Bacillales exhibited higher relative abundance at S1, particularly in roots (73.3%), compared to the top (3.7%) and bottom (3.1%) of shoots. Their abundance declined progressively from S1 to S3 but showed a rapid resurgence at the S4 stage, with relative abundances increasing notably at the shoot tops (42%), roots (29.1%), and shoot bottoms (5.6%). For fungal taxa, Saccharomycetales and Capnodiales were dominant.

### Microbial co-occurrence network and keystone taxa

Within each plant compartment, microbial interkingdom networks changed markedly across the four developmental stages (Supplementary Fig. 3). Moso roots exhibited higher network complexity than shoot tops and shoot bottoms (Fig. 2). The complexity and natural connectivity of endophytic microbial networks were higher at dormancy-breaking (S2) and gradually decreased during subsequent rapid growth phases. However, microbial communities at the shoot tops exhibited the highest network complexity during dormancy (S1), with the highest average degree (48.77) and clustering index (0.813).

OTUs belonging to Proteobacteria and Ascomycota showed stronger connections with OTUs in roots (Fig. 2). At shoot tops, OTUs belonging to Firmicutes and Ascomycota exhibited higher connectivity with other OTUs within the microbial network (Fig. 2).

### Core endophytic microbiota

Although the microbiome of Moso bamboo exhibited dynamic changes, core microbiome members remained prevalent in all plant compartment despite external influences. To elucidate the core microbiome of Moso bamboo, we analysed the temporal dynamics of microbial communities from roots, shoot tops and shoot bottoms, focusing on specific overlaps and indicator taxa of bacteria and fungi (Fig. 3, Supplementary Fig. 4). A total of 951 bacterial OTUs and 868 fungal OTUs were identified across the three plant compartment. Despite the overall community shifts, certain microbial taxa remained stable over time, forming a persistent core microbiome (Fig. 3). The core bacterial communities across different compartments were highly similar, predominantly consisting of *Cedecea*, *Pantoea*, *Serratia*, *Pseudomonas*, and *Bacillus*. Notably, *Paenibacillus*, *Ralstonia*, and *Lysinibacillus* also remained stable throughout all developmental stages in Moso roots. In contrast, fungal communities exhibited fewer stable taxa in roots over time. Only a limited number of genera, such as *Candida*, *Malassezia*, *Aspergillus*, *Penicillium*, and *Trichoderma*, persisting across different developmental stages.

### Variation of phytohormone and its correlation with microbial communities

LC-MS analysis revealed distinct spatial and temporal patterns in hormone distribution across Moso shoot compartments (Supplementary Fig. 5-8). The indole-3-pyruvic acid (IPA) levels were consistently highest in roots and increased markedly during development, peaking at 35 ng/g FW in S4. GA$_1$ and GA$_4$ predominated among gibberellins, with root concentrations peaking at S2. In shoot tops, trans-Zeatin (tZR) also exhibited peak levels during the S2 and S4 stages. During dormancy, ABA accumulated in shoot tops (501.57 ng/g FW) and bottoms (443.89 ng/g FW), with a significant reduction after dormancy. Conversely, JA and SA concentrations increased gradually with shoot development, with JA reaching its highest levels in roots.

Mantel test (a correlation analysis between distance matrices) analysis revealed that fungal communities across all Moso bamboo compartments were significantly correlated with phytohormones ($P < 0.001$). The general correlation between fungi and hormones across all tissue types suggested a broader effect of fungal communities on bamboo physiology. There was no significant correlation between hormones and bacterial communities in shoot tops and bottoms ($P > 0.05$), but there was a strong correlation between hormones and bacteria in roots ($P < 0.001$; Fig. 4).

### Dynamic changes in the expression of hormone-related genes in Moso shoot roots

To examine gene expression differences we used read counts from Moso bamboo roots at four developmental stages. We compared the expression profiles of roots across development stages and characterised 12,107 differentially expressed genes (DEGs) that were significantly up- or down-regulated. PCA revealed a clear separation of samples in the different development stages, especially between S1 and S3 (Supplementary Fig. 9).

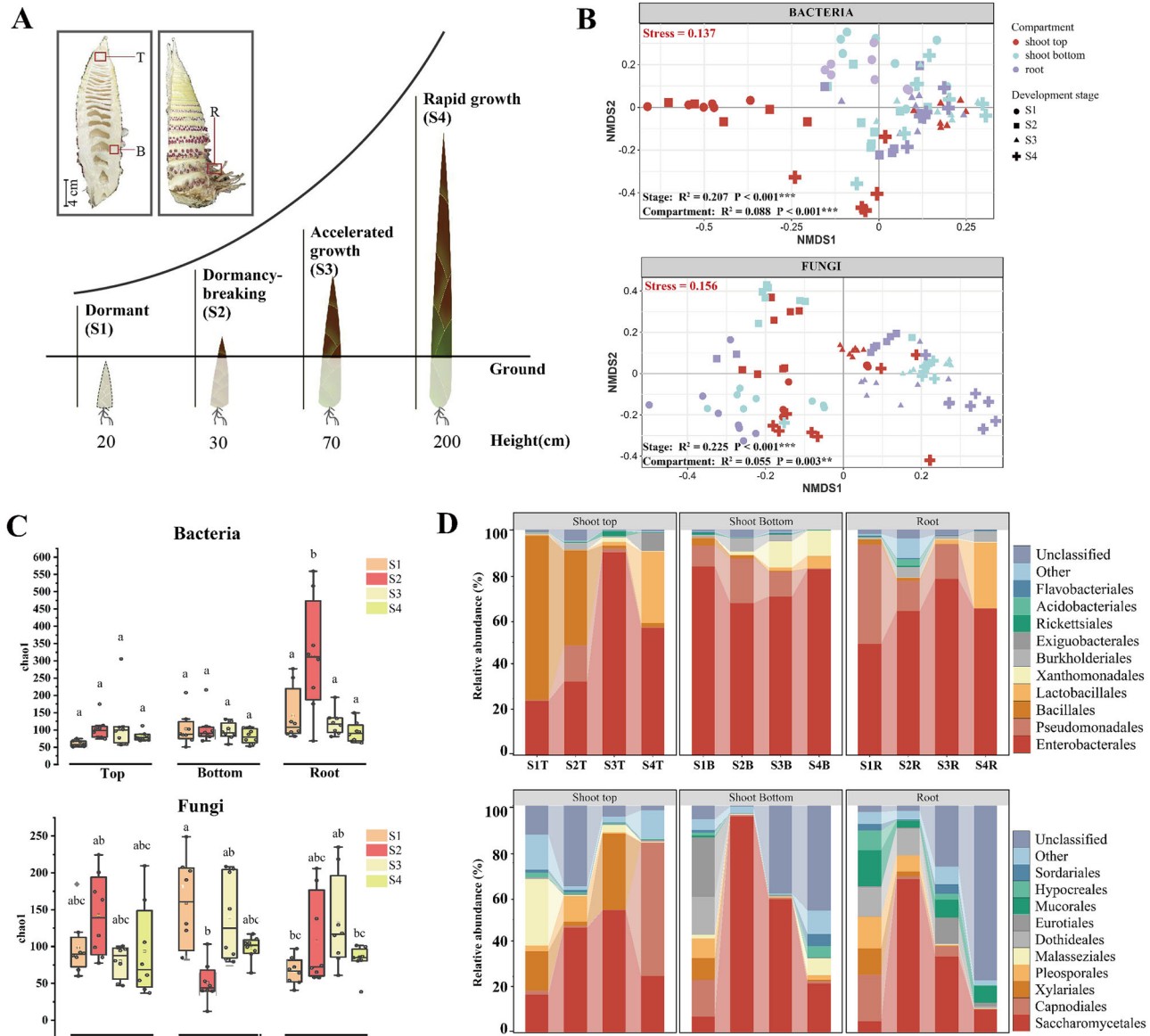

**Fig. 1 | Analysis of microbial diversity in different tissue parts of Moso bamboo shoots at different developmental stages. A** Schematic of four developmental stages of Moso bamboo shoots. Samples were collected from three plant compartments: shoot top (T), shoot bottom (B), and root (R). **B** Non-metric multidimensional scaling (NMDS) plots based on Bray-Curtis distances showing bacterial and fungal β-diversity across developmental stages and compartments. Stress values are indicated (bacteria: 0.137; fungi: 0.156), both < 0.2, reflecting reliable ordinations. Permutational multivariate analysis of variance (PERMANOVA) analysis indicated significant effects of both developmental stage and compartment (***$P < 0.001$, **$P < 0.01$). Each point represents one biological replicate ($n = 8$ per compartment per stage). **C** Boxplots of α-diversity (Chao1 index) for bacterial (top) and fungal

(bottom) communities across developmental stages and compartments. Different letters indicate significant differences ($P < 0.05$, one-way ANOVA with Tukey's HSD test). Data represent mean ± s.d. ($n = 8$ biologically independent samples per group). The line within each box represents the median, the box edges represent the interquartile range, and whiskers denote the minimum and maximum values. **D** Stacked bar charts showing the relative abundance of bacterial and fungal orders in different compartments and developmental stages. Relative abundances were calculated from sequencing data ($n = 8$ biologically independent samples per group). Error bars represent the standard deviation (SD) of $n = 8$ biologically independent samples.

Functional enrichment analysis showed that the major differential genes in the root system were enriched in pathways including the phenylpropane biosynthesis, starch sucrose metabolism, and phytohormone signalling (Supplementary Figs. 12-13).

We focused our analysis on the biosynthesis and signaling pathways of phytohormones, as well as the expression patterns of key related genes (Fig. 5). We observed that several genes encoding TAA1 were significantly upregulated during root development, consistent with the accumulation pattern of IPA (Fig. 5A). These results suggest that the roots actively synthesizes growth-promotion hormone via the IPA pathway during the rapid growth stage.

GA$_1$ and GA$_4$ showed significant accumulation in roots during the S2 stage. This coincided with the upregulation of PH02Gene48911, which encodes a key enzyme in the gibberellin synthesis pathway (Fig. 5D). Similarly, IPT3, which encodes Isopentenyltransferase-3 (IPT3) in the cytokinin biosynthesis pathway, was highly expressed during the dormancy-broken period, and the expression of CYP235A increased with Moso shoot growth (Fig. 5B).

Genes involved in the early steps of JA biosynthesis, such as lipoxygenases (LOXs), were highly expressed during the S2 stage. JA signaling, including allene oxide synthase (AOS), allene oxide cyclase (AOC), and oxophytodienoate reductase 7 (OPR7), were predominantly expressed during

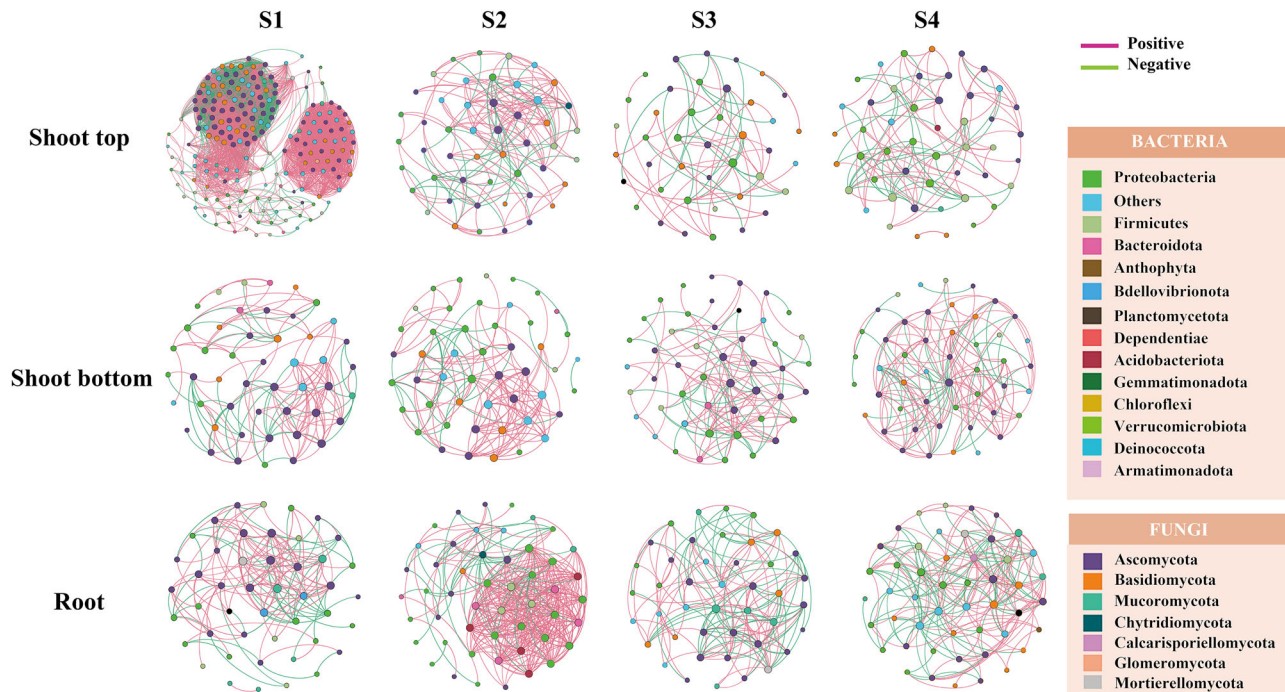

**Fig. 2 | Co-occurrence network analysis of bacteria and fungi in Moso shoots.** Microbial networks were constructed for three compartments (Shoot Top, Shoot Bottom, Root) at four developmental stages (S1: dormancy, S2: dormancy-breaking, S3: accelerated growth, S4: rapid growth). Each network was based on 8 biological replicates per group (*n* = 8). Each node represents a bacterial or fungal OTU, colored by taxonomic class as indicated in the legend. Edges represent robust and significant pairwise correlations calculated using Spearman's correlation ($|\rho| > 0.6$, false discovery rate [FDR]-adjusted $P < 0.05$). Positive correlations are shown as solid lines and negative correlations as dashed lines. Network topological properties (average degree, clustering coefficient, density, and modularity) are summarized in Table S1.

the S1 and S2 periods. Additionally, zeaxanthin epoxidase (ZEP) genes involved in ABA biosynthesis were significantly downregulated during the S1 and S2 periods.

To analyze phytohormone interactions, we constructed a network of differentially expressed hormone signaling genes (Supplementary Fig. 14). Among the fifteen most highly interconnected genes in the network (hub genes), nine are involved in the growth hormone signaling pathway, with seven belonging to the AUX/IAA family (Table S2).

### Correlation among phytohormone content, related gene expression and microbial communities in Moso shoot root

We considered correlations between 158 differentially expressed genes enriched for development stages and 89 microbial taxa (collapsed at the genus or final characterisation level and filtered by 0.1% relative abundance, cf. "Methods").

Twelve bacterial genus and one fungal genus were identified as members of the core microbial community. Among them, *Ralstonia*, *Paenibacillus* and *Lysinibacillus* showed high connectivity (degree > 10) in the network (Fig. 6, Table S3). The three core microbial taxa were mainly positively correlated with genes involved in auxin and cytokinin signaling pathways, particularly AUX/IAA and ARR-A genes. We further defined "network hubs" as nodes in the network with high degree (> 30) and strong centrality (> 0.3), and found a network hub, PH02Gene01141, which encodes the AUX/IAA protein. *Paenibacillus* and several other endophytes exhibited strong positive correlations with AUX/IAA genes (*P* < 0.01). In addition, *Paenibacillus* was also significantly positively correlated with genes encoding SAUR and ARR-A proteins.

### Discussion

Intensive selection pressures in plant interior compartments are driven by the host immune system as well as physical and biochemical barriers[16,17]. Distinct differences between rhizosphere and endosphere microbial communities have been widely reported across various plant species, including maize[18], rice[19], and *Arabidopsis*[20]. In our study, we observed that bacteria α-diversities was extremely low in S1 (Fig. 1C). This finding indicates a strong immune exclusion effect that limits microbial colonization[21]. However, microbial diversity increased significantly during the rapid growth stage (S3) of Moso bamboo (Fig. 1C). This suggests that Moso bamboo prioritizes resource allocation to growth at the expense of immune responses during the rapid growth stage (S3), thereby weakening its microbial selection pressure[22,23].

Positive and negative interactions of microbial co-occurrence networks are primarily considered potential cooperation and competition[24]. To better understand how internal plant structures influence microbial community dynamics, we examined the network complexity of endophytic microbes during different stages. The results indicate that plant compartment exerts a strong and direct influence on the complexity of endophytic microbial networks during S1 (Fig. 2). Notably, the shoot tops exhibited particularly high network complexity (Fig. 2). Shoot tops are dominated by meristematic tissues and represent a nutrient-poor environment[12,25]. Similar observations have been made in other nutrient-poor systems, where microbial communities form tightly linked networks to enhance survival under resource-limited conditions[26–28].

Meristems are characterized by strong immune barriers and tightly regulated hormonal signaling, making them relatively resistant to microbial invasion[29]. Therefore, endophytes capable of colonizing these regions are likely to possess specialized adaptive or stealth mechanisms, such as the expression of low-immunogenicity molecules or the secretion of immune-suppressive metabolites[30]. We found that the composition and structure of microbial community colonizing shoot tops differed significantly compared to the root (Figs. 1 and 2). The relative abundance of Bacillales at shoot top was much higher than that of other microorganisms at S1 stage (Fig. 1D). Recent studies suggest that an endophytic *bacillus subtilis* strain could evade plant defenses by masking self-produced flagellin with subtilomycin[23]. Zhang et al.[31] found that *B. velezensis* SQR9 is able to colonize plant roots in the presence of oxidative stress. In parallel, the fungal order Malasseziales, a

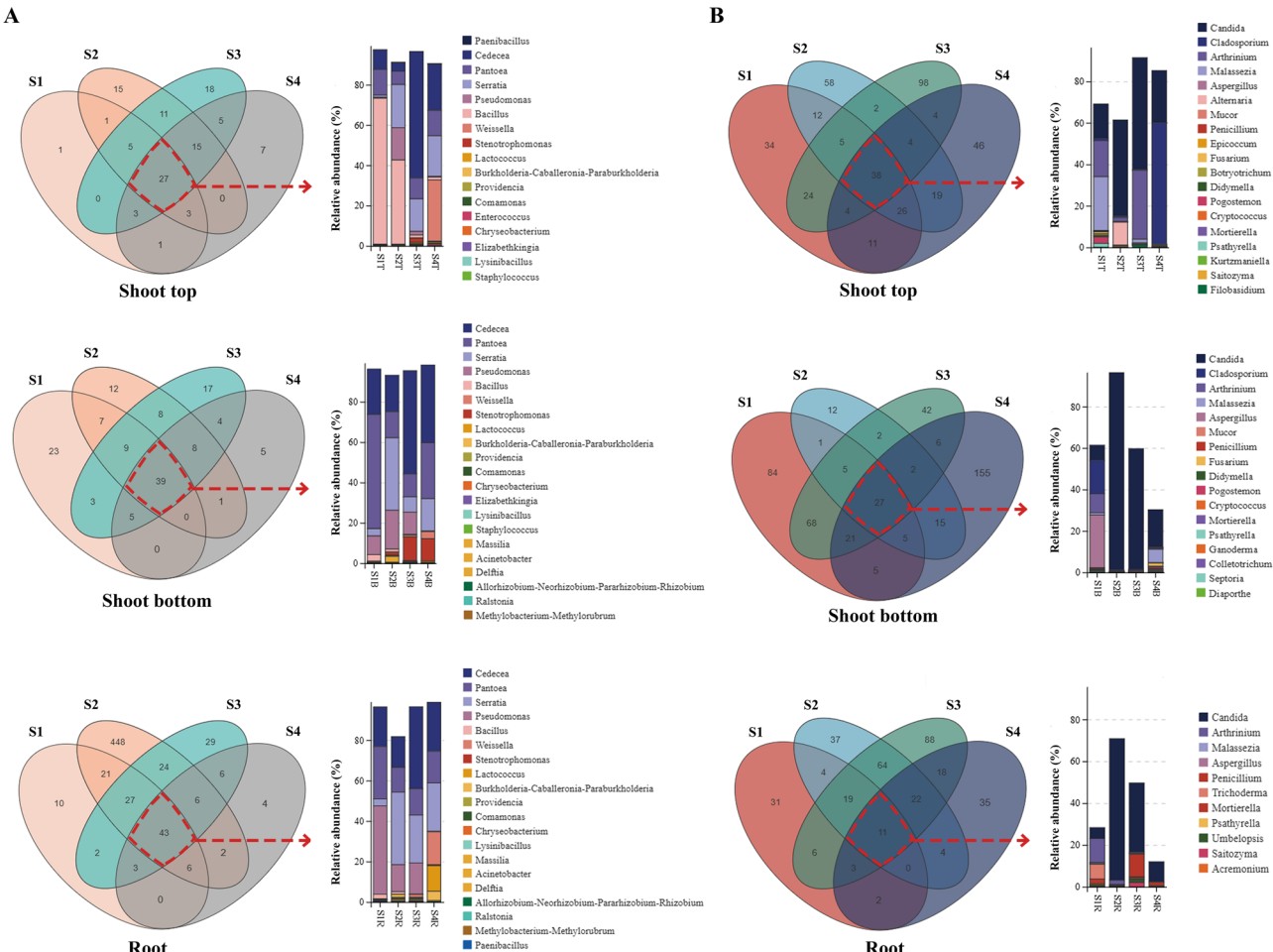

**Fig. 3 | Identification of core bacterial and fungal genera across different compartments and developmental stages of Moso bamboo. A** Venn diagrams and relative abundance bar plots of shared and unique bacterial genera in the top (T), bottom (B), and root (R) compartments at four developmental stages (S1 to S4).

**B** Corresponding analysis for fungal genera. Venn diagrams were generated based on OTU presence/absence across biological replicates (*n* = 8 per group). Stacked bar charts show the mean relative abundances of core genera (≥1% in at least one group) based on 16S rRNA (bacteria) and ITS (fungi) sequencing data.

group of lipid-dependent basidiomycetes typically found on plant surfaces or nutrient-poor tissues[32], was exclusively abundant in shoot meristems (Fig. 1D).

The co-occurrence of Bacillales and Malasseziales in this immune-restrictive, nutrient-limited niche suggests that both groups may possess specialized traits enabling persistence under strong plant defense and resource constraints, potentially through immune evasion, stress tolerance, or efficient utilization of available lipids and other limited resources. Such selective enrichment of immune-evasive and metabolically versatile taxa may provide functional advantages to the host, ensuring microbial stability in meristematic tissues and ultimately supporting the rapid growth of Moso bamboo shoots.

We identified core microbial taxa in shoot tops, shoot bottoms, and roots across developmental stages (Fig. 3). Gammaproteobacteria and Bacilli were the dominant bacterial orders, with Gammaproteobacteria showing the highest enrichment. Genera such as Cedecea, Pantoea, Serratia, and Pseudomonas were consistently present in internal compartments[33,34]. Bacilli, including Bacillus, Paenibacillus, and Lysinibacillus, were also stable members. These bacteria are known to fix atmospheric nitrogen, produce plant growth-promoting hormones, and enhance host adaptability[34]. Their persistence across compartments suggests a stable bacterial core that contributes to nutrient supply and growth regulation in Moso bamboo.

Fungal communities in roots contained fewer stable taxa. Only Candida, Malassezia, Aspergillus, Penicillium, and Trichoderma persisted across stages. Trichoderma produces phytohormone analogues and

antifungal metabolites that promote root growth and pathogen defense[35]. *Penicillium* and *Aspergillus* are efficient decomposers that utilize diverse carbon sources from root exudates, facilitating nutrient cycling[36]. Malassezia, although typically associated with animals, may exploit lipid-rich niches in meristematic tissues[32,37].

Together, these findings reveal that Moso bamboo harbors a compartment- and stage-independent core of bacterial and fungal taxa with distinct functional traits. The selection for microbes capable of nitrogen fixation, hormone production, or specialized nutrient utilization suggests strong host filtering for beneficial partners. Such stable microbial consortia may provide functional advantages that help sustain the extraordinary growth rate of Moso bamboo shoots.

As the primary interface between plants and soil, the root system is a critical zone where microbial activity and plant responses are highly dynamic[4,38,39]. Previous studies have demonstrated that successful root colonization by endophytes often requires microbial secretion of hormone or hormone-like compounds[37,40], which are essential for modulating root architecture and promoting plant growth. In our study, we found strong correlations between the abundance of root-associated fungal and bacterial taxa and the concentrations of key phytohormones in Moso bamboo roots (Fig. 4). These findings suggest that microbial communities in the root may influence or respond phytohormone signaling dynamics[15,30].

Therefore, we collected transcriptomic data from Moso bamboo roots at different developmental stages. Significant differential gene expression was observed in the roots during the development of Moso bamboo shoots

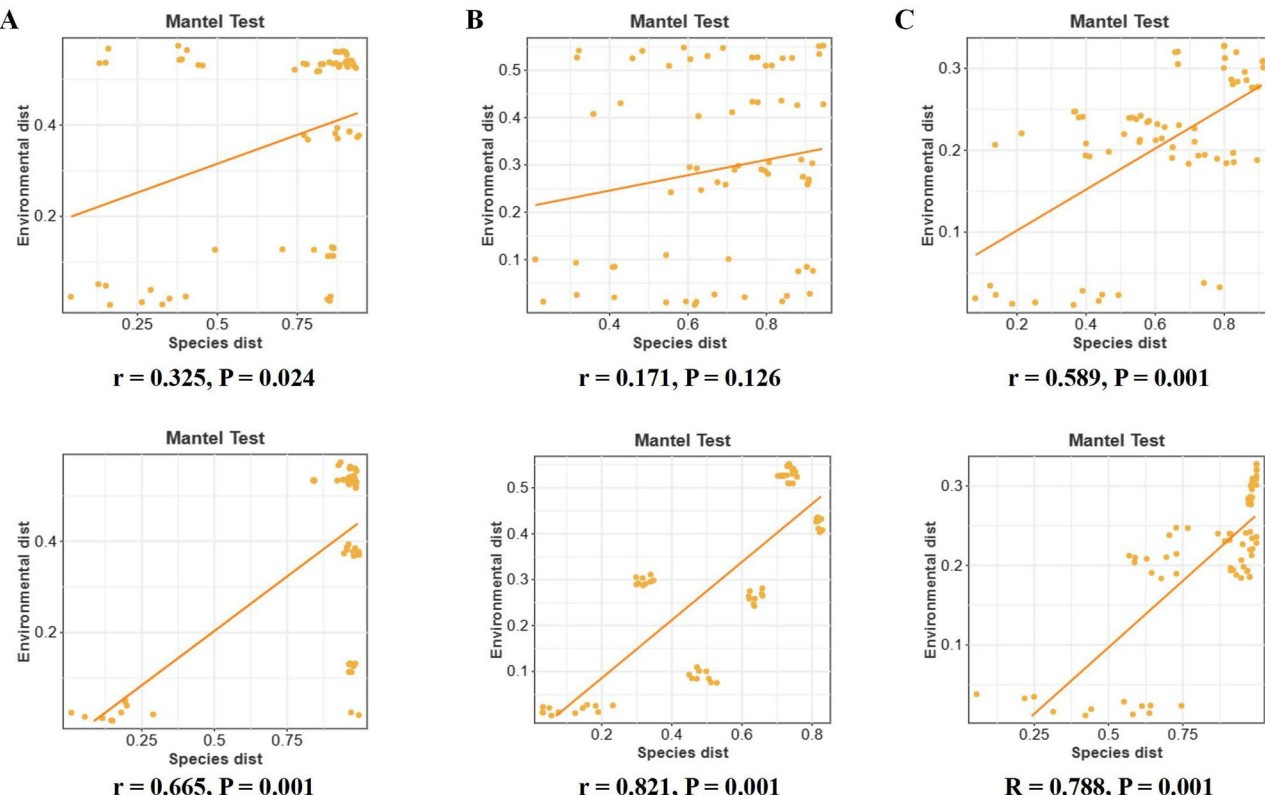

**Fig. 4 | Mantel test analysis showing the correlation between microbial community structure and phytohormone profiles in different compartments of Moso bamboo.** Mantel tests were used to assess the correlation between microbial community dissimilarity and phytohormone profile dissimilarity across three plant compartments: **A** shoot top, **B** shoot bottom, and **C** root. Each scatter plot shows the relationship between species distance (x-axis) and environmental distance based on hormone profiles (y-axis). Orange lines represent linear regression fits. Mantel test correlation coefficients (r) and corresponding *P*-values are shown for each compartment and microbial group.

(Supplementary Figs. 9-12). Genes related to the phenylpropanoid biosynthesis pathway were significantly upregulated during the S2 stage (Supplementary Fig. 13). Previous studies have shown that secondary metabolites derived from the phenylpropanoid biosynthesis pathway can be secreted by plant roots into the rhizosphere soil, promoting microbial at the root-soil interface[41,42]. This indicates that during the S2 stage, the roots of Moso shoots begin to secrete a large number of secondary metabolites, potentially attracting microbial communities.

Beyond the marked alterations in the phenylpropanoid biosynthesis pathway, significant changes were also observed in genes related to phytohormone biosynthesis and signal transduction (Supplementary Fig. 13). Among these, auxin signaling genes exhibited consistently high expression throughout the growth of Moso bamboo shoots and showed greater connectivity in the gene co-expression network compared to other hormone-related genes (Supplementary Fig. 14), highlighting their central regulatory role. Notably, the AUX/IAA gene family (early auxin-responsive transcriptional repressors that regulate root and shoot development) was identified as a key regulator of root development[43]. Twelve AUX/IAA genes were significantly upregulated during the S2 stage, suggesting that their high abundance may be required for auxin polar transport and cell division during early shoot elongation (Fig. 5).

In contrast, genes from the SAUR family, which are associated with cell elongation, were expressed at low levels during early development (S1-S2) and became highly upregulated in later stages, coinciding with the shift from cell division to elongation. This temporal expression pattern suggests that SAUR genes play a central role in the rapid cell expansion that drives the extraordinary shoot elongation of Moso bamboo[44]. Consistent with this view, genome-wide analyses in bamboo have shown that auxin-responsive genes, including SAUR and AUX/IAA, exhibit strong developmental stage-specific expression and are tightly linked to internode elongation. Similar developmental stage-dependent regulation of AUX/IAA and SAUR genes has been observed in rice and *Arabidopsis*. In *Oryza sativa*, overexpression of SAUR39 leads to reduced free IAA levels and impaired auxin transport, thereby limiting shoot and root elongation, which underscores its regulatory role during specific growth stages[45,46]. Together, these findings indicate that the stage-specific activation of AUX/IAA and SAUR gene families underpins the rapid root and shoot growth of Moso bamboo.

Microbial communities affect root development by secreting bioactive SMs that interact with auxin signaling components, such as TIR1, AUX/IAA, and ARFs[47–50]. For example, Trichoderma secretes the metabolite 2-aminoacetophenone (2-AA), which enhances auxin signaling and polar transport, potentially through the regulation of AUX/IAA gene activity[47]. Consistently, Fu et al. reported that beneficial bacteria promote plant growth by producing indole-3-acetic acid (IAA) and modulating auxin signaling components, thereby reshaping root system architecture.

Interestingly, our study found that AUX/IAA family genes exhibit significant correlations with several key microbial taxa within the plant-microbe-hormone network in Moso bamboo roots (Fig. 6). Notably, the plant growth-promoting bacterium *Paenibacillus* shows a strong association with the PH02Gene01142 gene. Previous studies have demonstrated that *Paenibacillus* can regulate maize root architecture by modulating auxin signaling and related metabolic pathways, thereby alleviating nutrient deficiencies[51].

These findings suggest a potential role for Paenibacillus in influencing bamboo growth through the modulation of AUX/IAA activity via secondary metabolites. Moreover, microbial regulation of auxin signaling is often integrated with cytokinin and ABA pathways, which could help explain how endophytes fine-tune the sequential activation of AUX/IAA

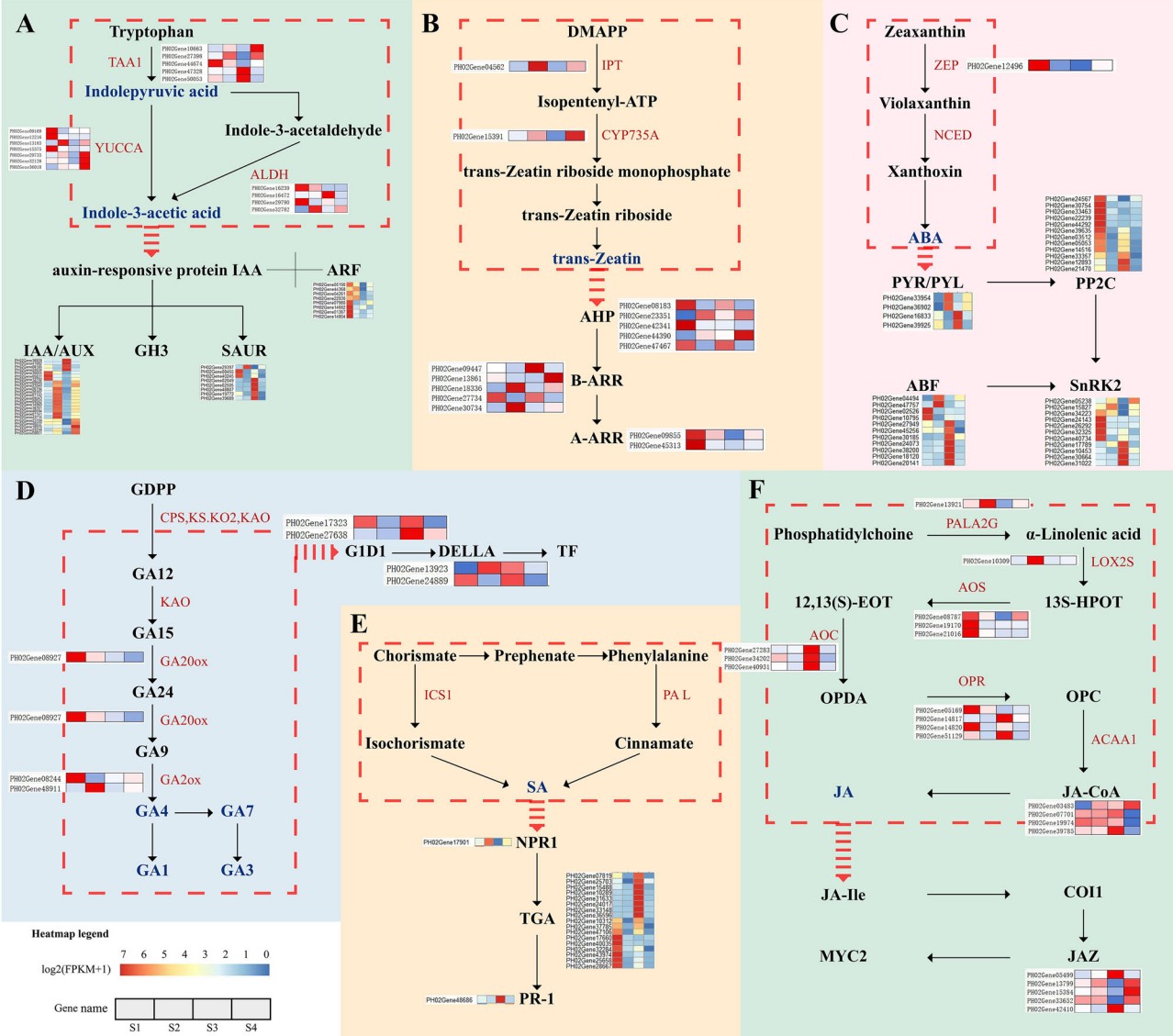

**Fig. 5 | Expression analysis of genes involved in phytohormone biosynthesis and signal transduction in Moso bamboo shoots.** Differentially expressed genes (DEGs) related to six phytohormone pathways are show - auxins (**A**), trans-zeatin (**B**) abscisic acid (**C**) gibberellins (**D**), salicylic acid (**E**) and jasmonic acid (**F**). Enzymes encoded by DEGs are highlighted in red. Heatmaps display log₂(FPKM + 1) expression values of each gene across four developmental stages (S1 to S4, from left to right). Each row represents a gene, and each column represents the mean expression of 3 biological replicates per stage (*n* = 3). DEGs were identified using DESeq2 (FDR-adjusted $P < 0.05$, $|\log_2\mathrm{FC}| \geq 1$).

and SAUR genes to drive bamboo shoot elongation[49,52]. It should be noted that these associations are correlative and do not prove direct causality. Future work involving metabolite identification and functional validation (e.g., exogenous application of candidate metabolites or gene knockout studies) will be essential to establish the causal role of endophytes in regulating hormone pathways.

This study provides multi-omics evidence, to the best of our knowledge, that endophytic microbes contribute to the exceptional growth of Moso bamboo by reprogramming hormone signaling. We demonstrated that: (i) microbial diversity and network complexity were compartment- and stage-dependent, with shoot tops during dormancy showing the highest complexity (average degree = 48.77, clustering index = 0.813), while root diversity increased markedly from S1 to S3; (ii) root-associated microbial shifts were tightly coordinated with hormone profiles, and transcriptome analysis identified 153 hormone-related genes, including stage-specific activation of AUX/IAA and SAUR families; and (iii) a plant-microbe-hormone interaction network highlighted *Paenibacillus* and other endophytes as potential modulators of auxin pathways driving rapid shoot

elongation. Together, these findings support a conceptual model (Fig. 7) in which immune filtering, endophytic colonization, and metabolite-mediated regulation of hormone signaling are dynamically coordinated across developmental stages. During dormancy (S1), strong immune barriers restrict microbial entry, whereas during growth stages (S2-S4), endophyte-derived metabolites may interact with AUX/IAA and SAUR signaling to accelerate cell expansion and shoot elongation. Looking ahead, identifying the specific microbial metabolites and their molecular targets will be essential to establish causal links and to harness endophytes for improving growth and stress resilience in fast-growing perennial species.

## Methods
### Plant materials
Based on the growth dynamics of Moso bamboo (*Phyllostachys edulis*), we defined four developmental stages: dormant (S1, shoots ~20 cm in height), dormancy-breaking (S2, shoots ~30 cm), accelerated growth (S3, shoots ~70 cm), and rapid growth (S4, shoots ~200 cm above ground). Shoot samples were collected from a managed bamboo forest in Wuxing District,

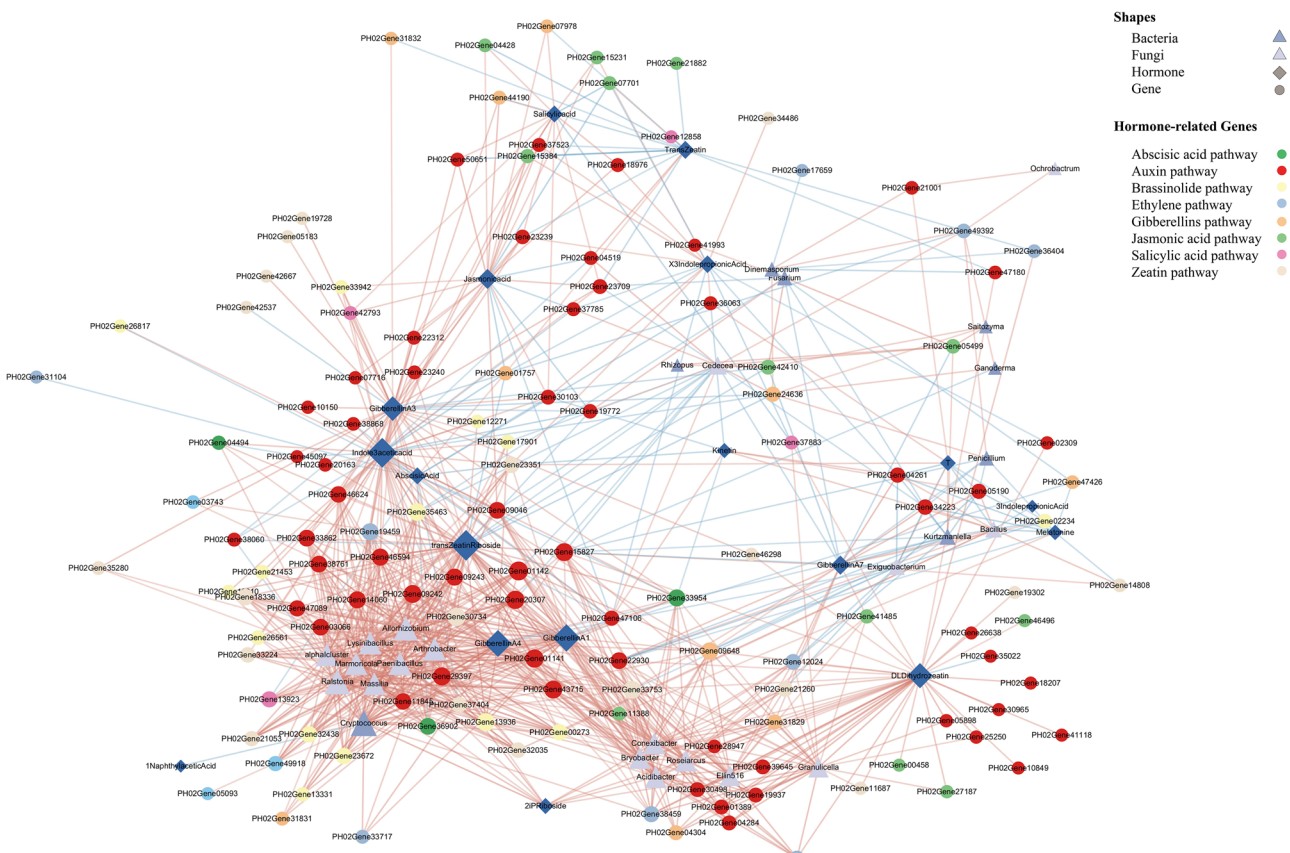

**Fig. 6 | Gene-microbe-hormone network.** This network depicts significant correlations ($|r| > 0.7$, $P < 0.01$) among microbial genera, phytohormones, and hormone-related genes identified in root samples. Nodes represent genes (circles), microbial taxa (diamonds), and hormones (triangles). Circle node colors correspond to different phytohormone metabolic or signaling pathways (auxin, abscisic acid, gibberellin, etc.) as indicated in the legend. Network constructed from $n = 3$ biologically independent root samples per stage; multiple testing controlled by FDR.

Huzhou City, Zhejiang Province, China (30°48'N, 119°59'E). This region has a northern subtropical monsoon climate, with four distinct seasons, an average annual temperature of 15.8 °C, average precipitation of ~1200 mm, and a frost-free period of ~230 days. Samples were collected on January 4 (S1), February 24 (S2), March 1 (S3), and April 27 (S4), 2023. Sample collection complied with local regulations and did not require specific permission.

A fixed 20 m × 20 m quadrat was established in the bamboo forest to minimize environmental variation. At each stage, eight individual shoots of similar height and thickness, free from mechanical damage or disease, were randomly selected within the quadrat to ensure comparability among replicates. Each shoot was dissected into three anatomical compartments: shoot top (apical bud and upper internodes), shoot bottom (basal culm section), and root system. Tissue blocks (~1 cm³) were excised from each compartment; more fragile tissues (e.g., apical buds) were sampled from the lower part to avoid breakage. In total, 96 samples were obtained (4 stages × 3 compartments × 8 replicates). All fresh tissues were immediately frozen in liquid nitrogen on site, transported to the laboratory on dry ice, and stored at −80 °C until further use. All samples were flash-frozen in liquid nitrogen and stored at −80 °C until further processing.

All 96 samples were subjected to microbiome profiling (amplicon sequencing of bacterial 16S rRNA and fungal ITS regions). For phytohormone profiling, a subset of 36 samples (4 developmental stages × 3 compartments × 3 replicates) was analyzed using LC-MS/MS. For transcriptomic analysis, only root tissues were selected, with 3 replicates per developmental stage (S1-S4), yielding a total of 12 RNA-seq libraries. This design enabled an integrative comparison of microbial communities, hormone dynamics, and gene expression across shoot compartments and developmental stages.

## Microbial DNA extraction and sequencing
Plant tissues were first rinsed under sterile distilled water and then surface-sterilized by sequential immersion in 75% ethanol (1 min) and 1% sodium hypochlorite (5 min), followed by 10 washes with sterile water. To confirm sterilization efficacy, the final rinse water was plated on LB agar and incubated at 30 °C for 2-3 days. Absence of colony growth was used as the criterion for successful surface sterilization.

Microbial DNA was extracted from the sterilized plant tissues using the HiPure Soil DNA Kit (Magen, China), which is suitable for low-biomass samples such as plant endospheres. DNA concentration and purity were evaluated with a NanoDrop 2000 spectrophotometer (Thermo Fisher Scientific, USA) and Qubit 3.0 fluorometer (Thermo Fisher Scientific, USA).

For bacterial community profiling, the V5-V7 region of the 16S rRNA gene was amplified using the primer pair 799 F (5′-AACMGGATTAGA-TACCCKG-3′) and 1193 R (5′-ACGTCATCCCCACCTTCC-3′). Fungal community profiling was performed by amplifying the ITS1 region using the primer pair ITS1_F_KYO2 (5′-TAGAGGAAGTAAAAGTCGTAA-3′) and ITS86R (5′-TTCAAAGATTCGATGATTCAC-3′), yielding an amplicon of approximately 366 bp. All PCR reactions were conducted using high-fidelity polymerase (New England Biolabs, USA). Sequencing libraries were purified and quantified before being pooled and sequenced using the Illumina MiSeq platform (2 × 300 bp paired-end reads).

Demultiplexed sequences were quality-filtered, trimmed, and clustered into operational taxonomic units (OTUs) at 97% identity using QIIME (v1.9.1). Taxonomic annotation was performed using the SILVA database (v138) for 16S sequences and the UNITE database (v8.3) for ITS sequences.

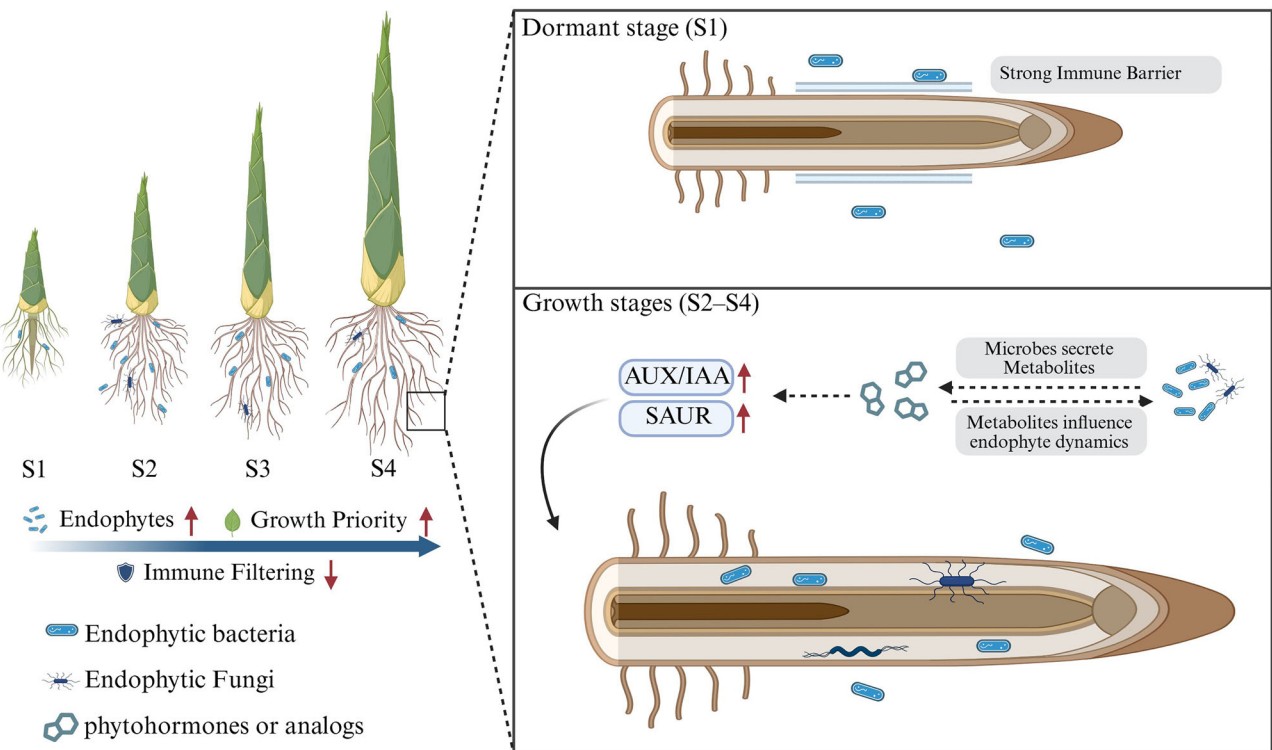

**Fig. 7 | Conceptual model of microbe-hormone interactions underlying rapid shoot growth of Moso bamboo.** The schematic integrates multi-omics findings across developmental stages. Left: Bamboo shoots from dormancy (S1) to rapid growth (S4), showing trends in immune filtering, endophyte abundance, and growth priority. Right, top: At S1, strong immune barriers restrict microbial entry, resulting in low endophytic diversity. Right, bottom: During growth stages (S2-S4), reduced immune filtering facilitates endophyte colonization. Endophytic bacteria and fungi secrete metabolites, which interact with hormone signaling and influence endophyte dynamics. AUX/IAA and SAUR gene families are upregulated, promoting cell expansion and shoot elongation. Dashed arrows indicate correlative or hypothesized interactions based on current data. Created with BioRender.com.

## Total RNA extraction, library preparation and transcriptome sequencing

Total RNA was extracted from the root tissues of Moso bamboo shoots (three biological replicates per developmental stage). The concentration and purity of the extracted RNA were determined using an ultra-micro spectrophotometer, Nanodrop2000 (manufactured by Thermo Fisher Scientific, USA). RNA integrity was evaluated by agarose gel electrophoresis and further assessed using an Agilent 5300 system (Agilent Technologies, USA) to determine the RNA Quality Number (RQN). A single library was required to have a total of 1 µg of RNA, a concentration ≥ 30 ng/µL, an RQN > 6.5, and an OD260/280 between 1.8 and 2.2. Poly(A) + RNA was isolated using oligo(dT) beads, and cDNA libraries were constructed with the NEBNext Ultra RNA Library Prep Kit for Illumina (New England Biolabs, USA) according to the manufacturer's instructions. The cDNA libraries were prepared using random primers and later sequenced on the Illumina NovaSeq X Plus platform. Clean reads were mapped to the Phyllostachys edulis reference genome available in BambooGDB, a bamboo genome database with functional annotation and analysis tool.[53] Gene expression levels were quantified using StringTie v2.1.4 and normalized to FPKM values for downstream differential expression analysis.

## Metabolite measurements

Plant tissues were thawed at 4 °C and approximately 100 mg fresh weight of each sample was transferred into a 10 mL centrifuge tube. Metabolites were extracted with 5 mL of pre-cooled extraction buffer (methanol:water:formic acid = 15:4:1, v/v/v, supplemented with 0.5% BHT as antioxidant). The mixture was vortexed for 1 min, sonicated for 30 min, and incubated at −40 °C for 60 min. After centrifugation (12000 rpm, 10 min, 4 °C), the supernatant was collected for solid-phase extraction (SPE). The SPE column was preconditioned with 3 mL water and 3 mL methanol, and sequentially eluted with 3 mL water and 10% methanol. The eluate was evaporated to dryness under nitrogen flow and reconstituted in 0.6 mL of 80% methanol (v/v). The solution was vortexed, centrifuged (12,000 rpm, 10 min, 4 °C), and the supernatant was subjected to LC–MS/MS analysis.

Targeted phytohormone analysis was performed on an AB SCIEX QTRAP 5500 mass spectrometer (Applied Biosystems, Foster City, CA, USA) equipped with an electrospray ionization (ESI) source, operating in multiple reaction monitoring (MRM) mode. Chromatographic separation was achieved using a Waters Acquity HSS T3 column (1.8 µm, 2.1 × 100 mm) with mobile phases consisting of (A) water containing 0.05% formic acid and (B) acetonitrile containing 0.05% formic acid. The flow rate was 0.3 mL/min under a linear gradient program: 0-1 min, 5% B; 1–9 min, 5-95% B; 9–12 min, 95% B; 12–12.1 min, 95-5% B; 12.1–15 min, 5% B for re-equilibration.

For quantification, calibration curves were constructed with authentic standards of major phytohormones (IAA, ABA, GA, SA, JA, cytokinins, etc.) over a concentration range of 0.1–100 ng/mL. All calibration curves exhibited excellent linearity with $R^2 > 0.99$. Stable isotope-labeled internal standards (e.g., D5-IAA, D6-ABA) were spiked into all samples to correct for matrix effects and monitor recovery rates. Representative total ion chromatograms (TICs), calibration curves are provided in Supplementary Figs. S16-S18.

Data acquisition and peak integration were performed using Multi-Quant software (AB SCIEX).

## Statistics and reproducibility

Alpha diversity indices (Chao1, ACE index, Shannon index, etc.) were calculated in QIIME (version 1.9.1). For Beta diversity analyses, the jaccard-based algorithm for determining differences in species community composition between samples was performed using the Vegan package (version 2.5.3)[45] based on OTUs and species abundance tables. The R language ggplot2

package (version 2.2.1) was used to plot NMDS (non-metric multi-dimensional scaling) analyses. In addition, for permutational multivariate analysis of variance (permutational multivariate analysis of variance, PERMANOVA) for beta distances the Vegan package (version 2.5.3) was used.

Venn analysis was performed using the VennDiagram package for R (version 1.6.16) to analyse OTUs of endemic species shared between groups. Biomarker species screening was performed using LEfSe software (version 1.0). Redundancy analysis (RDA, Redundancy analysis), mantel test was performed using R language Vegan package (version 2.5.3) to clarify the effect of environmental factors on community composition. Spearson correlation coefficients between environmental factors and species were calculated using the R language psych package (version 1.8.4).

Microbial interaction networks were constructed based on pairwise correlations between OTUs. Spearman correlation coefficients were calculated using the R package psych (version 1.8.4), and only robust correlations ($|\rho| > 0.6$, FDR-adjusted $P < 0.05$) were retained. The resulting networks were visualized in Cytoscape (version 3.3.0). The cutoff ($|\rho| > 0.6$, FDR-adjusted $P < 0.05$) was chosen following commonly applied standards in microbial co-occurrence network studies to ensure robustness and interpretability. Network topological properties, including number of nodes, number of edges, average degree, average weighted degree, density, modularity, average clustering coefficient, and average path length, were calculated to quantify the structural characteristics of microbial co-occurrence networks (Table S1).

DESeq2 (version 4.1.2) was used to analyze differential gene expression analyses using gene-level read count information. Differentially expressed genes (DEG) were screened. KEGG PATHWAY enrichment analysis was performed using the scipy (version 1.5.2) package in Python.

Correlation-based weighted networks were constructed by integrating hormone-related genes, phytohormone concentrations, and microbial OTUs. Pairwise Spearman's correlations were calculated using the R package psych (version 1.8.4). Correlations with $|\rho| > 0.6$ and FDR-adjusted $P < 0.05$ were retained. Unlike unweighted networks, edges in the final network were assigned weights corresponding to correlation coefficients, allowing representation of interaction strength. The weighted networks were visualized and analyzed in Cytoscape (version 3.3.0), where nodes represent genes, hormones, or microbial taxa, and edges represent significant correlations.

All boxplots and bar graphs show mean ± s.d. unless otherwise stated. Exact $P$ values, $n$ values, and the type of test are provided in figure legends. Each experiment was independently repeated at least three times, and the number of biological replicates ($n$) for each dataset is indicated in the corresponding figure legends or Methods subsections. Biological replicates refer to independent plant individuals collected from separate bamboo shoots, whereas technical replicates denote repeated measurements of the same sample.

## Reporting summary

Further information on research design is available in the Nature Portfolio Reporting Summary linked to this article.

## Data availability

The sequencing data generated in this study have been deposited in the NCBI Sequence Read Archive (SRA) under the following BioProject accession numbers: PRJNA1168663 and PRJNA1168528 (amplicon sequencing data for 16S rRNA and ITS), and PRJNA1168675 (root transcriptome data). Source data underlying all graphs and charts are provided in Supplementary Data 1 (Supplementary Data 1.xlsx).

## Code availability

No custom code was generated or used in this study. All analyses were conducted using publicly available software (QIIME v1.9.1; R v4.1.2 with vegan, ggplot2, psych; Cytoscape v3.3.0; DESeq2 v4.1.2; etc.).

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

## Acknowledgements

This research was supported by the Vanuatu Bamboo Development Project funded by the Global Cooperation Center, United Nations South-South Cooperation Office. This work was additionally supported by the Zhejiang Forestry Science and Technology Project (Project No. [2024] TS 17).

## Author contributions

Aoshun Zhao (A.Z.) and Xingcui Ding (X.D.) conceived the study and designed the overall research framework. Manchang Huang (M.H.), Aoshun Zhao (A.Z.) and Yingjie Cheng (Y.C.) performed the experiments, conducted sampling, and generated the multi-omics datasets. Aoshun Zhao (A.Z.) carried out data processing, statistical analyses, and visualization. Qiaoling Li (Q.L.), Anke Wang (A.W.), Yufang Bi (Y.B.) and Xuhua Du (X.D.2) contributed to data interpretation and manuscript revision. Xingcui Ding (X.D.) and Xuhua Du (X.D.2) supervised the project and acquired funding. All authors read and approved the final manuscript.

## Competing interests

The authors declare no competing interests.
