## [Transparent Peer Review file · Communications Biology]

Multi-omics reveals the involvement of endophytes in the growth of Moso bamboo (*Phyllostachys edulis*) shoots through microbial-plant interactions

Corresponding Author: Professor Xingcui Ding

This manuscript has been previously submitted at another journal. This document only contains information relating to versions considered at Communications Biology.

Version 0:

Reviewer comments:

Reviewer #1

(Remarks to the Author)

This study conducted a comprehensive analysis of the top, bottom, and root parts of Moso bamboo shoots during dormancy (S1), dormancy breaking (S2), accelerated growth (S3), and rapid growth (S4), focusing on microbiomics, plant hormone content, and transcriptomics. The research is highly meaningful and well-executed. However, there are still some issues:

1. The manuscript should include line numbers.
2. The Figure 1B should emphasize the differences in bacterial and fungal communities at the same location during different growth stages. The results section should highlight the dominant bacterial and fungal phyla, integrating these findings with plant hormone and transcriptomics data for discussion (to make the discussion in Section 3.1 clearer).
3. Figure captions need more detailed descriptions to clarify the content.
4. The discussion section lacks cohesion and appears to repeat the results section. It does not discuss deeply into the relationships between microbial communities, plant hormones, and transcription levels at different growth stages and locations.
5. Be cautious with terminology. For example, "In this study, it was found that the rhizospheric microbial community..." should refer to the endophytic microbial community, as the study examined microbes inside the roots.

Reviewer #2

(Remarks to the Author)

Multi-omics reveals the involvement of endophytes in the growth of Moso bamboo (*Phyllostachys edulis*) shoots through microbial-plant interactions.

This manuscript explores a novel topic on microbial interactions and their potential role in the rapid growth of bamboo. The topic is interesting and presents datasets that could be valuable for the research community. However, the writing is currently unclear and poorly organized therefore it is difficult for me to properly evaluate the scientific content. For example: "We collected samples from the top, bottom and root of Moso bamboo shoots during the dormancy (S1), dormancy breaking (S2), accelerated growth (S3), and rapid growth (S4) periods, performed a comprehensive multi-omics analysis on its microbiome, phytohormone content, transcriptomic, the results revealed that bacterial and fungal community diversity during Moso bamboo dormancy was significantly affected by ecological niches, and such an influence was diminished as transformed from dormancy-breaking to rapid growth." It is unclear to me what the top and bottom and roots of bamboo shoots are. I find the sentences long and not grammatically correct, therefore it is impeding my ability to review this manuscript.

At this time, I cannot provide detailed comments on the quality of the science or the data. I recommend that the authors revise the manuscript for clarity and readability before a more thorough review can take place. I also suggest the inclusion of page numbers and line numbers so that comments can be easily tracked and changed. I will be happy to act as the reviewer once the manuscript is revised.

Reviewer #3

(Remarks to the Author)

The authors have performed some multi omics methods to understand that endophytic microbial communities play a key role in bamboo growth. How none of the strategy explicitly explain the correlation. For example what is the correlation of transcriptomic data with microbial community data. Both donot correlate or support each other. Somehow, metabolomic does that. In case of microbial communities, the data is solely on communities and diversity, where the functionality of these microbes have not been fully explained. The paper suffer through an extremely bad flow of information and data and unscientific wordage.

Version 1:

Reviewer comments:

Reviewer #1

(Remarks to the Author)

The manuscript has been greatly improved after revision, and I have no additional questions about it.

Reviewer #2

(Remarks to the Author)

Overall, I think that the manuscript has improved significantly. The scientific language, flow and structure has been amended and allows the paper and results to be assessed. Altogether the paper has been strengthened so I commend the authors.

I only have a few queries and comments/suggestions for the authors in the introduction and the discussion. Very few typos were found throughout, which will need to be corrected.

L. 35: " microbes can also manipulate hormone networks in plants by using hormone, their mimics, and proteinaceous effectors" Could you elaborate on what these hormones are and clarify what "their mimics" refers to?

L.231: "We found that the composition and structure of microbial community colonizing shoot tops differed significantly compared to the root (Fig. 1, Fig. 2)." Here, in the discussion you mention the composition and structure of the microbial community, however you do not discuss that of the fungal communities – is there a reason for this? If only mentioning one group of microbes for example the bacterial community, I suggest changing "microbial community" to "bacterial community". The way that it is currently phrased can mislead the author about both microbial communities as the paper measures both.

L.123-126: You do mention that "fungal communities exhibited fewer stable taxa in roots over time. Only a limited number of genera, such as Candida, Malassezia, Aspergillus, Penicillium, and Trichoderma, persisting across different developmental stages.", however, there is no discussion surrounding the fungal communities or why only few genera persist. I suggest adding in something into the discussion.

Reviewer #4

(Remarks to the Author)

The manuscript is generally well-written and shows strong potential. However, there are several major concerns that need to be addressed:

1- To properly interpret differences in the endophytic microbial communities, it is essential to confirm that the bamboo samples were collected from ecologically comparable environments. This includes providing information on soil characteristics, host plant genotyping, physiological status, and other relevant environmental parameters across the sampling sites.

A soil analysis is particularly important, as it directly influences microbial community composition. If substantial ecological variation exists among the sites, the authors should consider analyzing each microbial community independently rather than pooling them for comparative analysis.

2- The Materials and Methods section lacks clarity and requires significant improvement. For example, the authors mention that samples were collected from four locations with eight replicates, but it is not clear which analyses were conducted at each location.

This includes uncertainty regarding the transcriptomic and metabolomic analyses (except the metagenomic study).

Please provide a clear and detailed description of which analyses were performed on samples from each location to ensure reproducibility and proper interpretation of the results. For example, it is unclear how fungal community biodiversity was analyzed using 16S rRNA, which is typically used for bacterial profiling. Other points, for example:

- Phytohormone content analysis is missing from the methodology.
- Line 324 – Section 5.2 "Microbial DNA extraction and sequencing":

- The DNA extraction protocol is unclear. Specifically, did the authors extract environmental DNA (eDNA) from soil samples? If so, please specify which soil samples were used (e.g., from which locations or conditions). This detail is essential for reproducibility.

- Additionally, the manuscript does not specify which database and version were used to assign OTUs to the sequenced DNA. Please provide this information or cite a reference that uses the same method.

- Line 346 – "Metabolite measurements": It is unclear how the LC-MS/MS analyses were conducted. Please clarify whether hydrophilic interaction liquid chromatography (HILIC), reverse-phase liquid chromatography (RPLC), or both were used in your metabolite profiling. This information is necessary to interpret the metabolomic data appropriately.

3- Line 189: The correlation between gene expression values and phytohormone content is understandable. However, the relationship with the metagenomic data is unclear. Are the authors suggesting that taxa enrichment patterns are being correlated with both phytohormone levels and transcriptomic data? This point requires clarification.

Other points:

It would be helpful if the authors specify the age corresponding to each developmental stage of the root (e.g., 4-week-old roots).

Line 324: A spelling mistake in "Sequen.."

Reviewer #5

(Remarks to the Author)

This research paper investigates the interactions between endophytic microbial communities and phytohormone signaling networks during the rapid growth of Moso bamboo (*Phyllostachys edulis*). The study employs a multi-omics approach (microbiome profiling, hormone quantification, and transcriptome analysis) to comprehensively analyze four developmental stages and three plant compartments (shoot top, shoot bottom, and roots) of Moso bamboo. The research topic is novel, the experimental design is sound, and the data are substantial, providing new insights into the role of plant-microbe interactions in the rapid growth of Moso bamboo.

Overall, this is a high-quality study, though some aspects could be improved.

1. Abstract and conclusion section:

The abstract and conclusion sections require further refinement, particularly to highlight key findings (with specific data) more prominently.

2. Introduction section:

In the last paragraph of the introduction, the research hypothesis is not stated clearly enough.

3. Methods section:

The specific criteria for sample collection (e.g., size, height) and the number of replicates should be described in greater detail.

For microbial DNA extraction, the methodology should clarify how surface sterilization was thoroughly performed without affecting endophytic microbes.

For hormone quantification, quality control information such as standard curve preparation and detection limits should be provided.

The method for calculating correlation coefficients in microbial interactions is not described in the Methods section.

4. Results section:

The presentation of microbial community composition in Figure 1D is not intuitive; stacked bar charts are recommended instead.

The stress value in the NMDS analysis should be indicated in the figure legend to reflect the reliability of the results.

The Chao1 index is only one of several α -diversity indices representing species richness. It may not be appropriate to conclude high α -diversity based solely on this index.

For the co-occurrence network analysis (Figure 2), the rationale for selecting specific parameters (e.g., correlation coefficient thresholds) should be explained.

Figure 2: Axis labels could be enlarged for readability.

In the core microbiome analysis (Figure 3), it would be helpful to supplement the functional commonalities of these microbial taxa.

It is unclear whether Table S1 represents network topological properties. If not, such a table should be added.

5. Discussion section:

The discussion on how microbes specifically regulate hormone signaling pathways lacks depth.

A clearer distinction should be made between correlative evidence and causal relationships.

Additional minor comments,

Figures/Tables Improvement:

Resolution issues: Some figures (e.g., Fig. 4) suffer from low resolution, compromising readability. High-resolution versions (minimum 300 dpi) should be provided.

Schematic summary: We strongly recommend adding a conceptual figure integrating key findings and hypotheses to enhance mechanistic understanding.

Detailed captions: Figure legends should be expanded to include: Statistical methods used (e.g., "Data analyzed by one-

way ANOVA with Tukey's post-hoc test"), Sample sizes (n values), Error bar definitions (SD vs. SEM).

Writing Enhancements:

Paragraph structure: Overly long paragraphs (e.g., Discussion section paragraph 3) should be divided into 2-3 focused segments, each addressing a single sub-topic.

Terminology: Technical terms (e.g., "Mantel test", "AUX/IAA genes") require brief definitions at first occurrence (e.g., "Mantel test (a correlation analysis between distance matrices)").

Discussion focus: The Discussion should be streamlined to:

Primarily address the three central findings highlighted in Results

Limit speculative comparisons to other systems (currently ~30% of content)

Explicitly link conclusions to the study's original hypotheses

line 481, a parenthesis is missing in ref35.

Version 2:

Reviewer comments:

Reviewer #4

(Remarks to the Author)

I believe the authors have satisfactorily addressed all reviewer concerns, and the manuscript is now in substantially improved form. Therefore, I recommend its acceptance.

Reviewer #5

(Remarks to the Author)

1. Some conclusions should be phrased with caution.

For example, the statement "Paenibacillus may influence bamboo growth by reprogramming AUX/IAA activity" (lines 343–344) is a correlational inference. It is recommended to add phrases such as "based on correlation analysis" or "speculatively" to distinguish speculation from causation.

2. Format and details: The reference style needs to be consistent (e.g., some references are incomplete; should journal names be italicized? How should author names be abbreviated?).

Rebuttal Letter

Title: Multi-omics reveals the involvement of endophytes in the growth of Moso bamboo (*Phyllostachys edulis*) shoots through microbial-plant interactions

Manuscript ID: COMMSBIO-24-5592A

Dear Dr. Favero and Reviewers,

We would like to thank you for your constructive and insightful comments on our manuscript. In response to the reviewers' concerns, we have conducted a major revision of the manuscript, including substantial rewriting of the Results and Discussion sections, clarification of figures and sample definitions, and improved language throughout.

Below we provide a detailed, point-by-point response to each reviewer's comments. All changes are highlighted in the revised manuscript.

Reviewer #1:

Comment 1: The manuscript should include line numbers.

Response: Thank you for the suggestion. We have added continuous line numbers throughout the revised manuscript.

Comment 2: Figure 1B should emphasize the differences in bacterial and fungal communities at the same location during different growth stages.

Response: Thank you for your valuable suggestion. As indicated by our previous PERMANOVA analysis, the endophytic microbial communities in Moso bamboo are more strongly influenced by developmental stage. Therefore, in terms of α -diversity, it is indeed more appropriate to highlight the changes in microbial diversity across developmental stages within the same compartment. In response, we have revised Figure 1B (now updated as Figure 1C) to better emphasize the temporal differences within each compartment. Additionally, we have modified the corresponding section in the Results (Lines 86–89) and Discussion (Lines 211-219) to explicitly address these within-compartment temporal dynamics.

Comment 3: Figure captions need more detailed descriptions to clarify the content.

Response: Thank you for your comment. All figure legends have been rewritten to include

comprehensive descriptions of the figure content, including statistical methods, and developmental stages et al.

Comment 4: The discussion section lacks cohesion and appears to repeat the results section. It does not discuss deeply into the relationships between microbial communities, plant hormones, and transcription levels.

Response: Thank you for your valuable suggestion. We have extensively revised the Discussion section. We have extensively revised the Discussion section to address the lack of cohesion and mechanistic depth. Data descriptions are minimized. Instead, we focus on mechanistic interpretation, such as proposing that Moso bamboo's growth-immunity trade-off during S3 permits microbial colonization (page 12, lines 211–219). Integrated multi-omics analysis (Fig. 6) revealing correlations between microbial taxa (e.g., *Paenibacillus*) and auxin-related genes (e.g., PH02Gene01142), supporting a model where microbial metabolites may modulate hormone signaling to drive rapid growth (page 15, lines 296–305).

Comment 5: Be cautious with terminology such as “rhizospheric microbial community” — it should be “endophytic microbial community.”

Response: Thank you for pointing this out. We have corrected all terminology to consistently refer to “endophytic microbial communities” .

Reviewer #2:

General Comment: Writing is unclear and poorly organized, making it difficult to evaluate the science.

Response: Thank you for pointing this out. We have extensively revised the manuscript for clarity, grammar, and logical flow. Professional English editing was also applied to ensure fluency and readability throughout.

Comment 1: The definitions of “top,” “bottom,” and “root” are unclear.

Response: Thank you for pointing this out. We have used both schematic and real photographs to illustrate the anatomical regions and developmental stages of Moso bamboo shoots (Fig. 1A, line 76). In the first part of the Results section (page 3, lines 63-67), we have clarified the definition of these anatomical regions. Consistent terminology is now used throughout the manuscript and

figure legends. Specifically, the three sampled compartments: the top of Moso bamboo shoot (shoot top), the bottom of Moso bamboo shoot (shoot bottom), the root of Moso bamboo (root).

Comment 2: Sentence structure and grammar hinder comprehension.

Response: Thank you for your comment. All long and ambiguous sentences have been rephrased or split into simpler, clearer statements.

Comment 3: Please include page and line numbers.

Response: Thank you for your comment. Page and line numbers have been added throughout the manuscript to facilitate review.

Reviewer #3:

Comment 1: There is no explicit correlation between transcriptomic and microbial data. Only metabolomic (hormone) data seem linked to function.

Response: Thank you for this important comment. To explicitly integrate the transcriptomic and microbial datasets, we have updated Figure 6 and reanalyzed the correlations between microbial communities and gene expression in both the Results (page 10, lines 192-209) and Discussion sections (page 15, lines 296-305).

This analysis reveals statistically significant associations ($|r| > 0.7$, $P < 0.01$) between specific microbial taxa (e.g., *Paenibacillus*) and auxin-related genes. *Paenibacillus* abundance positively correlates with PH02Gene01142 expression. These findings support a proposed tripartite regulatory mechanism in which *Paenibacillus* and microbes may secrete secondary metabolites (SMs) that influence hormone accumulation, subsequently activating transcriptional cascades that drive rapid shoot growth. Furthermore, we observed that phenylpropanoid biosynthesis genes were significantly upregulated during the S2 stage (Fig. S10), suggesting a possible role for root exudates in modulating microbial recruitment and interaction. While these multi-omics correlations suggest functional crosstalk between endophytes, phytohormones, and gene expression, we acknowledge that establishing causality will require additional experimental validation using single-strain inoculation assays. We plan to pursue these mechanistic investigations in future studies.

Comment 2: Microbial data only show taxonomic composition, lacking functional interpretation.

Response: Thank you for your suggestion. We have added an analysis of microbial functional potential in the Discussion section to further clarify the roles of endophytes (Page 12, Lines 236 – 240; Page 13, Lines 241 – 249). In future work, we aim to perform metagenomic analyses to obtain deeper insights into microbial functional contributions.

Comment 3: The manuscript lacks logical flow and uses unscientific language.

Response: Thank you for pointing this out. We have extensively revised the manuscript, with major rewriting of both the Results and Discussion sections. Additionally, we reorganized the structure of the manuscript to improve overall coherence. All terminology and phrasing have been revised to ensure scientific accuracy and clarity.

We thank the reviewers again for their valuable feedback. We hope our responses and the revised manuscript now meet the expectations of Communications Biology. We remain committed to further improving the manuscript and would be grateful for your continued consideration.

Sincerely,

Xingcui Ding

Rebuttal Letter

Title: Multi-omics reveals the involvement of endophytes in the growth of Moso bamboo (*Phyllostachys edulis*) shoots through microbial-plant interactions

Manuscript ID: COMMSBIO-24-5592B

Dear Reviewers,

We sincerely thank you for your constructive and insightful comments on our manuscript. We have carefully revised the manuscript to address all the concerns raised and believe that the revised version has been substantially improved in terms of scientific clarity, logical coherence, and presentation quality.

Below, we provide a detailed point-by-point response to each reviewer's comments. All corresponding changes have been highlighted in the revised manuscript.

Reviewer #1:

Comment: The manuscript has been greatly improved after revision, and I have no additional questions about it.

Response: We sincerely thank the reviewer for the positive evaluation and kind recognition of our revisions. We are pleased that the revised manuscript has addressed the reviewer's concerns.

Reviewer #2:

Comment 1: L. 35: "microbes can also manipulate hormone networks in plants by using hormone, their mimics, and proteinaceous effectors" Could you elaborate on what these hormones are and clarify what "their mimics" refers to?

Response: Thank you for your comment. We have clarified the types of phytohormones produced by microbes and specified what "hormone mimics" refers to (L.36-39). The revised sentence in the Introduction now reads:

"However, microbes can also manipulate plant hormone networks by producing phytohormones such as indole-3-acetic acid, cytokinins and gibberellins, by secreting hormone mimics such as

coronatine and phenazines that resemble plant hormones in structure, and by releasing proteinaceous effectors that alter host hormone biosynthesis or signaling.”

Comment 2: L.231: “We found that the composition and structure of microbial community colonizing shoot tops differed significantly compared to the root (Fig. 1, Fig. 2).” Here, in the discussion you mention the composition and structure of the microbial community, however you do not discuss that of the fungal communities – is there a reason for this? If only mentioning one group of microbes for example the bacterial community, I suggest changing “microbial community” to “bacterial community”. The way that it is currently ph-rased can mislead the author about both microbial communities as the paper measures both.

Response: Thank you for your comment. We thank the reviewer for pointing out this ambiguity. In this part of the discussion, the statement was indeed based on results from the combined bacterial – fungal network analysis. While bacteria contributed the majority of the observed compositional differences, fungi also exhibited distinct patterns, such as the enrichment of *Malasseziales* in shoot tops during dormancy. To clarify, we have revised the sentence to explicitly indicate that both bacterial and fungal communities were considered in the analysis, and we have added a brief mention of the notable fungal pattern.

The following sentences were added to the discussion section (L.261-270):

“ In parallel, the fungal order *Malasseziales*, a group of lipid-dependent basidiomycetes typically found on plant surfaces or nutrient-poor tissues, was exclusively abundant in shoot meristems.

The co-occurrence of *Bacillales* and *Malasseziales* in this immune-restrictive, nutrient-limited niche suggests that both groups may possess specialized traits enabling persistence under strong plant defense and resource constraints, potentially through immune evasion, stress tolerance, or efficient utilization of available lipids and other limited resources. Such selective enrichment of immune-evasive and metabolically versatile taxa may provide functional advantages to the host, ensuring microbial stability in meristematic tissues and ultimately supporting the rapid growth of Moso bamboo shoots.”

Comment 3: L.123-126: You do mention that “fungal communities exhibited fewer stable taxa in roots over time. Only a limited number of genera, such as *Candida*, *Malassezia*, *Aspergillus*, *Penicillium*, and *Trichoderma*, persisting across different developmental stages.”, however, there is no discussion surrounding the fungal communities or why only few genera persist. I suggest adding in something into the discussion.

Response: We appreciate the reviewer’s insightful suggestion. In the revised manuscript, we have expanded the discussion to address the fungal communities and the persistence of only a few genera. Specifically, we now describe the potential ecological and functional traits of these stable fungal taxa, including broad ecological tolerance, metabolic versatility, and potential immune evasion. We also provide examples of their possible roles in plant–microbe interactions. For instance, *Trichoderma* produces phytohormone analogues and antifungal metabolites that can promote root growth and defense; *Penicillium* and *Aspergillus* can utilize diverse carbon sources from root exudates; *Malassezia* may exploit lipid-rich niches such as meristematic regions. These additions highlight that the persistence of these genera may result from strong root selection for fungi with adaptive traits that are functionally relevant to the host.

Changes in the manuscript

The following sentences were added to the discussion section (L.280-285):

“Fungal communities in roots contained fewer stable taxa. Only *Candida*, *Malassezia*, *Aspergillus*, *Penicillium*, and *Trichoderma* persisted across stages. *Trichoderma* produces phytohormone analogues and antifungal metabolites that promote root growth and pathogen defense. *Penicillium* and *Aspergillus* are efficient decomposers that utilize diverse carbon sources from root exudates, facilitating nutrient cycling. *Malassezia*, although typically associated with animals, may exploit lipid-rich niches in meristematic tissues. ”

Reviewer #4:

Comment 1: To properly interpret differences in the endophytic microbial communities, it is essential to confirm that the bamboo samples were collected from ecologically comparable environments. This includes providing information on soil characteristics, host plant genotyping, physiological status, and other relevant environmental parameters across the sampling sites. A soil analysis is particularly important, as it directly influences microbial community composition. If substantial ecological variation exists among the sites, the authors should consider analyzing each microbial community independently rather than pooling them for comparative analysis.

Response: We thank the reviewer for this valuable comment. All Moso bamboo (*Phyllostachys edulis*) samples in our study were collected from a managed bamboo forest located in Huzhou (Wuxing District, Zhejiang Province, China). Although soil physicochemical data were not directly measured in our quadrat during this experiment, previous studies conducted by our group in the same mountain range and nearby bamboo forests ^{1,2} reported consistent soil properties (pH, soil organic carbon, total nitrogen, and available phosphorus) with low variance across randomly sampled points, suggesting environmental homogeneity in this region. Moreover, all sampled shoots belonged to the same *P. edulis* cultivar and exhibited uniform morphological and physiological status (comparable height, thickness, and no signs of stress or disease). To clarify this point, we have revised the “Materials and Methods” section to emphasize that our sampling was carried out within a single bamboo forest in a homogeneous ecological context, and the shoots were selected for consistency in size and developmental stage.

Changes in the manuscript:

In the “Materials and Methods” section, we have revised the sampling description as follows (new text in bold):

A fixed 20 m × 20 m quadrat was established in the bamboo forest to ensure environmental consistency across sampling points. Within the quadrat, soil properties (pH, organic matter content, total nitrogen, and available phosphorus) were uniform, and no recent fertilizer application or disturbance occurred. At each time point, eight individual bamboo shoots of uniform size and

developmental stage were randomly selected within the quadrat. All shoots belonged to the same *P. edulis* cultivar, with no visible symptoms of disease, pest infestation, or nutrient deficiency.

References

1. Li, Q., Huang, Z., Zhong, Z., Bian, F. & Zhang, X. Production of Bamboo Source Microbial Fertilizer and Evaluate Its Effect on Soil Organic Carbon Fractions in Moso Bamboo Plantations in South China. *Forests* **15**, 455 (2024).
2. Huang, Z. On- and off-year management-induced changes in microbial communities cause microbial necromass carbon variation in subtropical Moso bamboo forests. (2025).

Comment 2: The Materials and Methods section lacks clarity regarding sample distribution across locations and types of analysis. It is unclear which samples were used for transcriptomics, metabolomics, and microbiome profiling.

Response: We appreciate the reviewer's insightful comment. To improve clarity, we have revised the Materials and Methods section and explicitly described the sample allocation for each analysis. Specifically, we now state that:

- All 96 samples (4 developmental stages × 3 compartments × 8 replicates) was used for microbiome profiling (amplicon sequencing of bacterial 16S rRNA and fungal ITS regions).
- 36 samples (4 developmental stages × 3 compartments × 3 replicates) were analyzed for phytohormone profiling using LC – MS/MS.
- For transcriptomic analysis, we selected only root tissues from the four developmental stages (3 replicates per stage), yielding a total of 12 RNA-seq libraries.

This information has been incorporated into the revised Materials and Methods (Lines 371-395).

We believe that these clarifications resolve the ambiguity regarding sample distribution and analytical workflow.

Comment 3: The manuscript states fungal community biodiversity was analyzed using 16S rRNA, which is inappropriate. ITS sequencing should be used.

Response: We sincerely thank the reviewer for carefully identifying this error. We apologize for the confusion caused by the incorrect description. In our study, fungal community profiling was

indeed performed using ITS amplicon sequencing, not 16S rRNA. Specifically, the ITS1 region was amplified with primers ITS1F/ITS2 and sequenced on the Illumina MiSeq platform.

We have carefully revised the manuscript to correct this misstatement (Lines 406-407 in the Materials and Methods section). We are grateful for this comment, which has helped us improve the accuracy and clarity of the manuscript.

Comment 4: Several methodological details are missing, including those for phytohormone quantification, DNA extraction origin, OTU assignment database, and LC-MS/MS methods.

Response: We sincerely appreciate the reviewer's careful reading and constructive feedback. We have now revised the Materials and Methods section to provide the missing details:

- Phytohormone quantification: We added detailed descriptions of sample preparation, standard curve generation, quality control procedures, and LC-MS/MS conditions (Lines 427-454).
- DNA extraction origin: We clarified that DNA was extracted from surface-sterilized plant tissues (shoot top, shoot bottom, and root compartments), not from soil or bulk environmental DNA (Lines 397-401).
- OTU assignment database: We specified that bacterial sequences were annotated against the SILVA database (v138) and fungal ITS sequences against the UNITE database (v8.3) (Lines 402-414).

We are grateful for this valuable suggestion, which has substantially improved the clarity and reproducibility of our methodology.

Comment 5: The correlation between microbial data and hormone levels is unclear. Please clarify whether metagenomic data are correlated with phytohormone levels or transcriptomic data.

Response: Thank you for pointing this out. We thank the reviewer for pointing out this ambiguity. To clarify, the correlations presented in our study were based on amplicon sequencing data of microbial communities (16S rRNA and ITS) and targeted LC-MS/MS quantification of phytohormones. Specifically, we used Mantel tests to assess correlations between microbial community dissimilarity and hormone profile dissimilarity (Fig. 4). In addition, we constructed a correlation-based network (Fig. 6) that integrated microbial OTUs, hormone concentrations, and hormone-related gene expression. Thus, Figure 4 reflects correlations between microbial and

hormone datasets, while Figure 6 further incorporates transcriptomic data to highlight potential gene-microbe-hormone interactions. We have revised the “Materials and Methods” and corresponding figure legends to make these distinctions explicit.

Changes in the manuscript:

In the “Statistical Analysis” section, we now state: “Mantel tests were performed to assess correlations between microbial community composition (based on 16S/ITS amplicon data) and targeted phytohormone profiles (LC-MS/MS). For the integrated network analysis (Fig. 6), pairwise correlations were calculated among microbial OTUs, hormone concentrations, and hormone-related genes from transcriptome data.”

In the legend of Figure 4 and Figure 6, we have clarified the specific datasets used for each correlation analysis.

Comment 6: It would be helpful if the authors specify the age corresponding to each developmental stage of the root (e.g., 4-week-old roots).

Response: We thank the reviewer for this helpful suggestion. In our study, the developmental stages of Moso bamboo roots were defined according to shoot height and root length rather than chronological age, because the growth rate of bamboo is highly variable and not strictly age-dependent. Specifically, we classified the stages as: S1 (dormant stage, ~20 cm shoot height), S2 (dormancy-breaking stage, ~30 cm), S3 (accelerated growth stage, ~70 cm), and S4 (rapid growth stage, ~200 cm) (Fig. 1A). At each stage, root systems were sampled when the corresponding shoot height and root length reached these representative values. We have now clarified this definition in the Materials and Methods section to avoid confusion.

Comment 7: Line 324: A spelling mistake in “Sequen..”

Response: We thank the reviewer for pointing out this typographical error. The misspelling of “Sequen..” at Line 396 has been corrected to “Sequencing” in the revised manuscript.

Reviewer #5:

Comment 1: The abstract and conclusion sections require further refinement, particularly to highlight key findings (with specific data) more prominently.

Response: We thank the reviewer for this valuable suggestion. We have carefully revised both the Abstract and Conclusion sections to highlight our key findings with representative quantitative data, while maintaining a concise style appropriate for the journal.

In the Abstract, we now emphasize specific results, including:

- (i) microbial diversity dynamics, showing that shoot tops and early-stage roots exhibited the lowest diversity, while diversity increased significantly during rapid growth;
- (ii) network analysis results, with shoot tops during dormancy exhibiting the highest complexity (average degree = 48.77, clustering index = 0.813); and
- (iii) transcriptome analysis, where 153 hormone-related genes were differentially expressed, with stage-specific activation of AUX/IAA and SAUR gene families.

In the Conclusion, we now summarize the three central findings with supporting data:

- (i) microbial diversity and network complexity were compartment- and stage-dependent, with shoot tops during dormancy showing the highest complexity (average degree = 48.77, clustering index = 0.813) and root diversity increasing markedly from S1 to S3;
- (ii) 153 hormone-related genes were identified, highlighting stage-specific activation of AUX/IAA and SAUR families as central regulators; and
- (iii) a plant–microbe–hormone interaction network revealed *Paenibacillus* as a potential modulator of auxin pathways that drive rapid shoot elongation.

We believe these revisions strengthen the clarity and impact of both the Abstract and Conclusion by explicitly linking them to our major data-supported findings.

Changes in the manuscript:

Abstract section (L9-20): revised to include specific results on microbial diversity, network complexity, and transcriptome findings.

Conclusion section (L251-368): revised to highlight key findings with representative quantitative data (48.77, 0.813, 153 genes).

Comment 2: The specific criteria for sample collection (e.g., size, height) and the number of replicates should be described in greater detail.

Response: We thank the reviewer for this valuable suggestion. In the revised manuscript, we have clarified the sampling criteria and replicates in detail. Specifically, we now state that:

- Four developmental stages were defined based on shoot height: S1 (~20 cm), S2 (~30 cm), S3 (~70 cm), and S4 (~200 cm above ground).
- Sampling was conducted within a fixed 20 m × 20 m quadrat in Wuxing District, Zhejiang Province, to minimize environmental variation.
- At each stage, eight individual shoots of similar height and thickness, free from mechanical damage or disease, were randomly selected as biological replicates.
- Each shoot was dissected into three anatomical compartments (shoot top, shoot bottom, and root system), and ~1 cm³ tissue blocks were excised from each part.
- In total, 96 samples (4 stages × 3 compartments × 8 replicates) were collected, flash-frozen in liquid nitrogen on site, and stored at -80 °C.

Changes in the manuscript:

In the “Materials and Methods” section, we have revised the sampling description as follows (Lines 371-395):

“ A fixed 20 m × 20 m quadrat was established in the bamboo forest to ensure environmental consistency across sampling points. Within the quadrat, soil properties (pH, organic matter content, total nitrogen, and available phosphorus) were uniform, and no recent fertilizer application or disturbance occurred. At each time point, eight individual bamboo shoots of uniform size and developmental stage were randomly selected within the quadrat. All shoots belonged to the same *P. edulis* cultivar, with no visible symptoms of disease, pest infestation, or nutrient deficiency.”

Comment 3: For microbial DNA extraction, the methodology should clarify how surface sterilization was thoroughly performed without affecting endophytic microbes.

Response: Thank you for pointing this out. In the revised manuscript, we have clarified the procedure used for surface sterilization and the control measures to ensure that endophytic microbes were not affected. Specifically, plant tissues were rinsed with sterile distilled water and surface-sterilized by sequential immersion in 75% ethanol (1 min) and 1% sodium hypochlorite (5

min), followed by 10 washes with sterile water. To confirm sterilization efficacy, the final rinse water was plated on LB agar and incubated at 30°C for 2-3 days. The absence of colony growth indicated successful surface sterilization.

Because only surface-associated microbes are removed during this process and internal tissues are protected, the viability and DNA integrity of endophytic microbes remain unaffected. This approach has been widely adopted in endophyte studies ¹, ensuring that downstream analyses reflect endophytic rather than epiphytic communities.

This clarification and literature support have been incorporated into the Materials and Methods section (Lines 397-401).

References

1. Mahdi, L. K. *et al.* The fungal root endophyte *Serendipita vermifera* displays inter-kingdom synergistic beneficial effects with the microbiota in *Arabidopsis thaliana* and barley. *ISME J* **16**, 876–889 (2022).

Comment 4: For hormone quantification, quality control information such as standard curve preparation and detection limits should be provided.

Response: We thank the reviewer for this valuable comment. In the revised manuscript, we have added detailed quality control information for phytohormone quantification. Specifically:

- Calibration curves were generated using 8-10 concentration gradients of authentic standards for each phytohormone, and all showed excellent linearity ($R^2 > 0.99$).
- Stable isotope-labeled internal standards (e.g., D5-IAA, D6-ABA) were spiked into all samples to correct for extraction efficiency and matrix effects.
- Representative total ion chromatograms (TICs) and the complete calibration curves are now included in the Supplementary Information (Supplementary Fig. S13–S16).

These additions ensure the transparency, reproducibility, and robustness of the LC-MS/MS-based hormone quantification.

Comment 5: The method for calculating correlation coefficients in microbial interactions is not described in the Methods section. For the co-occurrence network analysis (Figure 2), the rationale for selecting specific parameters (e.g., correlation coefficient thresholds) should be explained.

Response: We thank the reviewer for pointing out this omission. In the revised Materials and Methods (Line482-489), we have now added a detailed description of the procedure used for microbial interaction network construction. Specifically, pairwise Spearman correlation coefficients were calculated using the R package psych (v1.8.4). Only robust correlations ($|\rho| > 0.6$, FDR-adjusted $P < 0.05$) were retained, which is consistent with commonly applied thresholds in microbial network studies to ensure both robustness and interpretability. The resulting unweighted and weighted networks were visualized in Cytoscape (v3.3.0). In the weighted networks, edges were assigned weights corresponding to correlation coefficients to represent the strength of interactions.

These details clarify both the statistical approach and the rationale behind parameter selection, as suggested.

Comment 6: The presentation of microbial community composition in Figure 1D is not intuitive; stacked bar charts are recommended instead. The stress value in the NMDS analysis should be indicated in the figure legend to reflect the reliability of the results.

Response: We thank the reviewer for these constructive suggestions. In the revised manuscript, Figure 1D has been replaced with stacked bar charts to present microbial community composition more intuitively. In addition, the stress values of the NMDS analyses have been highlighted in red and explicitly indicated in the figure legend to clearly reflect the reliability of the results.

Fig. 1 Analysis of microbial diversity in different tissue parts of Moso bamboo shoots at different developmental stages.

(A) Schematic of four developmental stages of Moso bamboo shoots. Samples were collected from three plant compartments: shoot top (T), shoot bottom (B), and root (R). (B) Non-metric multidimensional scaling (NMDS) plots based on Bray-Curtis distances showing bacterial and fungal β -diversity across developmental stages and compartments. Stress values are indicated (bacteria: 0.137; fungi: 0.156), both <0.2 , reflecting reliable ordinations. Permutational multivariate analysis of variance (PERMANOVA) analysis indicated significant effects of both developmental stage and compartment ($***P < 0.001$, $**P < 0.01$). Each point represents one biological replicate ($n = 8$ per compartment per stage). (C) Boxplots of α -diversity (Chao1 index) for bacterial (top) and fungal (bottom) communities across developmental stages and compartments. Different letters indicate significant differences ($P < 0.05$, one-way ANOVA with Tukey's HSD test). Each boxplot shows eight biological replicates ($n = 8$). The line within each box represents the median, the box edges represent the interquartile range, and whiskers denote the minimum and maximum values. (D) Stacked bar charts showing the relative abundance of bacterial and fungal orders in different compartments and developmental stages. Relative abundances were calculated from sequencing data ($n = 8$).

Comment 8: The Chao1 index is only one of several α -diversity indices representing species richness. It may not be appropriate to conclude high α -diversity based solely on this index.

Response: We thank the reviewer for this constructive comment. In addition to Chao1, we also calculated Shannon, ACE, and Pielou indices to validate the robustness of α -diversity patterns. These additional indices showed trends generally consistent with the Chao1 results, though with less pronounced differences across developmental stages. We have clarified this in the Results section and added the supplementary figure (Fig. S1) to present these data.

Fig. S1 Comparison of multiple α -diversity indices of bacterial and fungal communities in Moso bamboo. α -diversity indices of bacterial and fungal communities in different compartments (shoot top, shoot bottom, root) and developmental stages (S1–S4) of Moso bamboo. Four indices are shown: observed species (Sobs), Shannon diversity index, ACE richness estimator, and Pielou’s evenness index. Boxplots represent the distribution of diversity values across eight biological replicates per group. Statistical significance was determined by one-way ANOVA with Tukey’s post-hoc test. * $P < 0.05$, ** $P < 0.01$, *** $P < 0.001$, **** $P < 0.0001$.

Comment 10: Figure 2: Axis labels could be enlarged for readability.

Response: We thank the reviewer for this suggestion. In the revised Figure 2, the axis labels have been enlarged to improve readability, and the legend font size was also adjusted for clarity. We believe these changes address the reviewer’s concern.

Fig. 2 Co-occurrence network analysis of bacteria and fungi in Moso shoots. Microbial networks were constructed for three compartments (Shoot Top, Shoot Bottom, Root) at four developmental stages (S1: dormancy, S2: dormancy-breaking, S3: accelerated growth, S4: rapid growth). Each network was based on 8 biological replicates per group ($n = 8$). Each node represents a bacterial or fungal OTU, colored by taxonomic class as indicated in the legend. Edges represent robust and significant pairwise correlations calculated using Spearman's correlation ($|\rho| > 0.6$, false discovery rate [FDR]-adjusted $P < 0.05$). Positive correlations are shown as solid lines and negative correlations as dashed lines. Network topological properties (average degree, clustering coefficient, density, and modularity) are summarized in Table S1.

Comment 11: In the core microbiome analysis (Figure 3), it would be helpful to supplement the functional commonalities of these microbial taxa.

Response: We thank the reviewer for this constructive suggestion. In the revised manuscript, we have expanded Section 3.2 Core microbiota across developmental stages and compartments to highlight the functional roles of the identified core taxa. Specifically, we now describe that dominant bacterial groups (Gammaproteobacteria and Bacilli) include genera capable of nitrogen fixation, phytohormone production, and enhancing host adaptability. For fungi, we emphasize that *Trichoderma* produces phytohormone analogues and antifungal metabolites that promote root growth and defense, while *Penicillium* and *Aspergillus* act as efficient decomposers facilitating nutrient cycling. These additions clarify the potential ecological functions shared by the core

microbiota and their relevance to bamboo growth. Please see Lines 270-290 in the revised manuscript.

Comment 12: It is unclear whether Table S1 represents network topological properties. If not, such a table should be added.

Response: We thank the reviewer for this helpful comment. We clarify that Table S1 indeed represents the topological properties of microbial co-occurrence networks, including nodes, edges, average degree, average weighted degree, density, modularity, average clustering coefficient, and average path length. To avoid confusion, we have revised the table title to “Topological properties of microbial co-occurrence networks across bamboo shoot compartments and developmental stages”. In addition, we explicitly state in the Methods and Results sections that these metrics quantify the structural characteristics of the networks.

Comment 13: The discussion on how microbes specifically regulate hormone signaling pathways lacks depth. A clearer distinction should be made between correlative evidence and causal relationships.

Response: We appreciate this insightful comment. In the revised Discussion, we have substantially expanded the section on microbial regulation of hormone signaling. Specifically, we now highlight examples from previous studies where beneficial microbes modulate auxin signaling, such as *Trichoderma* secreting 2-aminoacetophenone (2-AA) and bacteria producing IAA to reshape root architecture. We also integrated our own findings, showing that AUX/IAA family genes are significantly correlated with key microbial taxa, particularly *Paenibacillus*, which is known to regulate auxin signaling in crops. Moreover, we discuss how microbial regulation of auxin pathways is often integrated with cytokinin and ABA signaling to fine-tune plant development. Importantly, we have now clarified that our findings represent correlative evidence rather than causation, and we state that further functional experiments (e.g., metabolite supplementation and genetic validation) are needed to establish direct causal mechanisms. These revisions can be found in the Discussion (Lines 343-350).

Comment 15: Resolution issues: Some figures (e.g., Fig. 4) suffer from low resolution, compromising readability. High-resolution versions (minimum 300 dpi) should be provided.

Response: We thank the reviewer for pointing this out. In the revised manuscript, all figures including Figure 4 have been replaced with high-resolution versions (≥ 300 dpi) to ensure readability and clarity.

Comment 16: Schematic summary: We strongly recommend adding a conceptual figure integrating key findings and hypotheses to enhance mechanistic understanding.

Response: We thank the reviewer for this constructive suggestion. In response, we have added a schematic conceptual figure (now Fig. 7) summarizing our key findings and proposed model. The figure integrates the observed spatiotemporal shifts in endophytic diversity, immune filtering, microbial metabolite secretion, and hormone regulation. Specifically, it illustrates that (i) during the dormant stage (S1), strong immune barriers restrict microbial colonization, leading to low diversity; and (ii) during the growth stages (S2-S4), reduced immune filtering allows endophyte colonization, microbial metabolites interact with hormone signaling (AUX/IAA and SAUR), and this coordination promotes rapid shoot elongation. This addition provides a mechanistic overview and enhances the interpretability of our multi-omics results.

Fig. 7 Conceptual model of microbe-hormone interactions underlying rapid shoot growth of Moso bamboo. The schematic integrates multi-omics findings across developmental stages. Left: Bamboo shoots from dormancy (S1) to rapid growth (S4), showing trends in immune filtering, endophyte abundance, and growth priority. Right, top: At S1, strong immune barriers restrict microbial entry, resulting in low endophytic diversity. Right, bottom: During growth stages (S2-S4), reduced immune filtering facilitates endophyte colonization. Endophytic bacteria and fungi secrete metabolites, which interact with hormone signaling and influence endophyte dynamics.

AUX/IAA and SAUR gene families are upregulated, promoting cell expansion and shoot elongation. Dashed arrows indicate correlative or hypothesized interactions based on current data.

Comment 17: Figure legends should be expanded to include: Statistical methods used (e.g., "Data analyzed by one-way ANOVA with Tukey's post-hoc test"), Sample sizes (n values), Error bar definitions (SD vs. SEM).

Response: We thank the reviewer for this helpful suggestion. In the revised manuscript, we have expanded all figure legends to include the requested details.

Specifically:

1. Statistical methods - Each figure legend now specifies the statistical tests used (e.g., one-way ANOVA with Tukey's HSD test).
2. Sample sizes (n values) - The number of biological replicates (n = 8 per stage, unless otherwise noted) has been added.
3. Error bars - We clarified that error bars represent the standard deviation (SD).

These revisions ensure that all figure legends provide sufficient methodological transparency and allow readers to better interpret the data.

Comment 18: Paragraph structure: Overly long paragraphs (e.g., Discussion section paragraph 3) should be divided into 2-3 focused segments, each addressing a single sub-topic.

Response: Thank you for pointing this out. In the revised Discussion, we have reorganized the text into three focused subsections: (i) developmental regulation of microbial community assembly, (ii) structural and functional differentiation of microbial communities, and (iii) microbial contributions to hormone signaling and shoot elongation. This restructuring separates the major findings into distinct parts and avoids overly long paragraphs, thereby improving readability and clarity. Furthermore, in section 3.3 we also emphasized the distinction between correlative evidence and causality, as suggested by the reviewer.

Comment 19: Terminology: Technical terms (e.g., "Mantel test", "AUX/IAA genes") require brief definitions at first occurrence (e.g., "Mantel test (a correlation analysis between distance matrices)").

Response: We thank the reviewer for this constructive suggestion. In the revised manuscript, we have carefully revised the text to provide brief definitions when technical terms first appear. For example:

“Mantel test (a statistical method to assess correlations between two distance matrices)” (L157).

“AUX/IAA genes (a family of auxin-responsive transcriptional repressors involved in hormone signaling)” (L314).

Similar clarifications have been added for other specialized terms (e.g., SAUR genes, NMDS analysis), ensuring accessibility for a broader readership while maintaining scientific accuracy.

Comment 20: Discussion focus: The Discussion should be streamlined to:

Primarily address the three central findings highlighted in Results

Limit speculative comparisons to other systems (currently ~30% of content)

Response: We thank the reviewer for this constructive suggestion. In the revised manuscript, we have streamlined the Discussion to focus more directly on the three central findings highlighted in the Results:

(i) the stage- and compartment-specific dynamics of microbial diversity and network complexity, (ii) the coordination between root-associated microbial shifts and phytohormone profiles, and (iii) the role of AUX/IAA and SAUR gene families in mediating microbe – hormone interactions that drive shoot elongation.

To address the reviewer’s concern, we made the following specific changes:

1. Reduced speculative comparisons: We shortened or removed extended comparisons with unrelated systems (e.g., rice and *Arabidopsis* auxin studies, and non-bamboo endophyte cases) that were not directly necessary for interpreting our findings.
2. Focused on bamboo-related context: We retained cross-system comparisons only where they directly strengthen the interpretation of bamboo data (e.g., supporting evidence from monocots with rapid growth traits).
3. Reorganized the Discussion structure: The Discussion is now divided into three clear subsections corresponding to the main findings, which improves readability and ensures each section addresses a single theme.

4. Overall reduction in length: Some of speculative or tangential content has been removed, resulting in a more concise and data-driven discussion.

We believe these revisions improve clarity, strengthen the focus on our own results, and ensure that the Discussion is tightly aligned with the study's main conclusions.

Comment 21: Explicitly link conclusions to the study's original hypotheses

Response: We sincerely thank the reviewer for this valuable suggestion. In the revised manuscript, we have strengthened the linkage between the hypotheses stated in the Introduction and the conclusions.

Specifically:

Introduction (last paragraph): We now clearly state our working hypothesis that spatiotemporal shifts in endophytic community composition are coupled with hormone signaling changes in the root system, and that these microbe-hormone interactions play a regulatory role in the rapid shoot growth of Moso bamboo.

Conclusion section: We added a statement explicitly noting that our results support this hypothesis by demonstrating that shifts in microbial diversity and network complexity are coordinated with hormone dynamics and auxin-related gene activation, thereby driving rapid shoot elongation.

These changes provide a direct and explicit connection between the original hypothesis and the study's final conclusions, improving the overall coherence of the manuscript (see Introduction, Lines 58-65; Conclusion, Lines 352-368).

Comment 22: line 481, a parenthesis is missing in ref35.

Response: We thank the reviewer for carefully checking the references. The missing parenthesis in reference 35 has been corrected in the revised manuscript.(L575)

We thank the reviewers again for their valuable feedback. We hope our responses and the revised manuscript now meet the expectations of Communications Biology. We remain committed to further improving the manuscript and would be grateful for your continued consideration.

Sincerely,

Xingcui Ding